# CAUSAL EXPLANATIONS OF STRUCTURAL CAUSAL MODELS

## ABSTRACT

In explanatory interactive learning (XIL) the user queries the learner, then the learner explains its answer to the user and finally the loop repeats. XIL is attractive for two reasons, (1) the learner becomes better and (2) the user's trust increases. For both reasons to hold, the learner's explanations must be useful to the user and the user must be allowed to ask useful questions. Ideally, both questions and explanations should be grounded in a causal model since they avoid spurious fallacies. Ultimately, we seem to seek a causal variant of XIL. The question part on the user's end we believe to be solved since the user's mental model can provide the causal model. But how would the learner provide causal explanations? In this work we show that existing explanation methods are not guaranteed to be causal even when provided with a Structural Causal Model (SCM). Specifically, we use the popular, proclaimed causal explanation method CXPlain to illustrate how the generated explanations leave open the question of truly causal explanations. Thus as a step towards causal XIL, we propose a solution to the lack of causal explanations. We solve this problem by deriving from first principles an explanation method that makes full use of a given SCM, which we refer to as SC**E** (**E** standing for explanation). Since SCEs make use of structural information, any causal graph learner can now provide human-readable explanations. We conduct several experiments including a user study with 22 participants to investigate the virtue of SCE as causal explanations of SCMs.

## 1 INTRODUCTION

There has been an exponential rise in the use of machine learning, especially deep learning in several real-world applications such as medical image analysis (Ker et al., 2017), particle physics (Bourilkov, 2019), drug discovery (Chen et al., 2018) and cybersecurity (Xin et al., 2018) to name a few. While there have been several arguments that claim deep models are interpretable, the practical reality is much to the contrary. The very reason for the extraordinary discriminatory power of deep models (namely, their depth) is also the reason for their lack of interpretability. To alleviate this shortcoming, interpretable and explainable AI/ML (Chen et al., 2019; Molnar, 2020) has gained traction to explain algorithm predictions and thereby increase the trust in the deployed models. However, providing explanations to increase user trust is only part of the problem. Ultimately, explanations or interpretations (however one defines these otherwise ill-posed terms) are a means for humans to understand something—in this case the deployed AI model. Therefore, a closed feedback loop between user and model is necessary for both boosting trust through understanding/transparency and improving models robustness by exposing and correcting their shortcomings. The new paradigm of XIL (Teso & Kersting, 2019) offers exactly the described where a model can be "right or wrong for the right or wrong reasons" and depending on the specific scenario the user-model interaction will adapt (e.g. giving the right answer and a correction when being "wrong for the wrong reasons").

Now the question arises, what would constitute a good explanation inline with human reasoning? In their seminal book, Pearl & Mackenzie (2018) argue that causal reasoning is the most important factor for machines to achieve true human-level intelligence and ultimately constitutes the way humans reason. Several works in cognitive science are indeed in support of Pearl's counterfactual theory of causation as a great tool to capture important aspects of human reasoning (Gerstenberg et al., 2015; 2017) and thereby also how humans provide explanations (Lagnado et al., 2013). The authors in

Hofman et al. (2021) even argue that systems that are efficient in both causality and explanations are need of the hour. Questions of the form "What if?" and "Why?" have been shown to be used by children to learn and explore their external environment (Gopnik, 2012; Buchsbaum et al., 2012) and are essential for human survival (Byrne, 2016). These humane forms of causal inferences are part of the human mental model which can be defined as the illustration of one's thought process regarding the understanding of world dynamics (see also discussions in Simon (1961); Nersessian (1992); Chakraborti et al. (2017)). All these views make understanding and reasoning about causality an inherently important problem and suggest that what we truly seek is a causal variant of XIL in which the explanations of the model are grounded in a causal model and the user is also allowed to give feedback about causal facts.

While acknowledging the difficulty of the problem we address it pragmatically by leveraging qualitative (partial) knowledge on the SCM (Pearl, 2009) justifying the naming of our truly causal explanations (by construction) that we will be referring to as *Structural Causal **Explanation*** (SCE). The motivation behind this work is the observation that spurious associations in the training data inevitably leads to the failure of non-causal models and explanation methods. For example, an image classifier that learns on watermarked images will have high accuracy on the test data from the same distribution but it will be "right for the wrong reasons" and furthermore the "right" part is brittle as it will fail when moving out-of-distribution (Lapuschkin et al., 2019). In psychology this phenomenon of spurious association fallacy is known as "Clever Hans" behavior named after the 20th century Orlov Trotter horse Hans that was wrongly believed to be able to perform arithmetic (Pfungst, 1911). Some works such as (Stammer et al., 2021) moved beyond basic methods (like heat maps for image data) using a XIL setup to move beyond "Clever Hans" fallacies. Some other works purely on the explanation part made an effort in devising a "causal" explanation algorithm to avoid spuriousness (Schwab & Karlen, 2019), however, as we show in this work they still leave open the question of truly causal explanations–a gap that we fit. We provide a new, natural language expressible (thus, human understandable) explanation algorithm with SCE.

Overall, we make several contributions: **(I)** we devise from first principles a new algorithm (SCE) for computing explanations from SCM making them truly causal explanations by construction, **(II)** we showcase how SCE fixes several of the shortcomings of previous explainers, **(III)** we apply the SCE algorithm to several popular causal inference methods, **(IV)** we discuss using a synthetic toy data set how one could use SCE for improving model learning, and finally **(V)** we perform a survey with 22 participants to investigate the difference between user and algorithmic SCE.

We make our code repository publically available at: https://anonymous.4open.science/r/Structural-Causal-Explanations-D0E7/

## 2 BACKGROUND AND RELATED WORK

We briefly review key concepts from previous and related work to establish a high-level understanding of the basics needed for the discussion in this paper.

**Causality.** Following the Pearlian notion of Causality (Pearl, 2009), an SCM is defined as a 4-tuple $\mathcal{M} := \langle \mathbf{U}, \mathbf{V}, \mathcal{F}, P(\mathbf{U}) \rangle$ where the so-called structural equations (which are deterministic functions) $v_i \leftarrow f_i(\mathrm{pa}_i, u_i) \in \mathcal{F}$ assign values (denoted by lowercase letters) to the respective endogenous/system variables $V_i \in \mathbf{V}$ based on the values of their parents $\mathrm{Pa}_i \subseteq \mathbf{V} \setminus V_i$ and the values of some exogenous variables $\mathbf{U}_i \subseteq \mathbf{U}$ (sometimes also referred to as unmodelled or nature terms), and $P(\mathbf{U})$ denotes the probability function defined over $\mathbf{U}$. The SCM formalism comes with several interesting properties. They induce a causal graph $G$, they induce an observational/associational distribution over $\mathbf{V}$ (typical question "What is?", example "What does the symptoms tell us about the disease?"), and they can generate infinitely many interventional/hypothetical distributions (typical question "What if?", example "What if I take an aspirin, will my headache be cured?") and counterfactual/retrospective distributions (typical question "Why?", example "Was it the aspirin that cured my headache?") by using the *do*-operator which "overwrites" structural equations. Note that, opposed to the Markovian SCM discussed in for instance (Peters et al., 2017), the definition of $\mathcal{M}$ is semi-Markovian thus allowing for shared $U$ between the different $V_i$. Such a shared $U$ is also called *hidden confounder* since it is a common cause of at least two $V_i, V_j (i \neq j)$. Opposite to that, a "common" confounder would be a common cause from within $\mathbf{V}$. In the case of linear SCM, where the structural equations $f_i$ are linear in their arguments, we call the coefficients *"dependency terms"*

and they fully capture the causal effect of $V_i$ onto $V_j$ denoted as $\alpha_{i \to j}$, that is, how changing $V_i$ will change $V_j$. The depedency terms of non-linear SCM are generally more involved, so $\alpha_{i \to j}$ cannot be obtained by simply reading off coefficients in $f_i$. In most settings the causal effect, e.g. of a medical treatment onto the patient's recovery, is the sought quantity of interest. If interventions are admissible, then the average causal (or treatment) effect (ACE/ATE) within a binary system is defined as a difference in interventional distributions, that is $\alpha_{i \to j} := \mathbb{E}[V_j \mid do(V_i = 1)] - \mathbb{E}[V_j \mid do(V_i = 0)]$. A great deal of research in causality (especially for ML) is concerned with leveraging observational data to reason about causal relationships (also known as identification), which often times overlaps with the broader study of identifying the graphical structure underlying the data distribution. For example, Ke et al. (2019) made use of data from the first two levels of Pearl's Causal Hierarchy (PCH), namely observational and interventional, to update their graph estimate while using masked neural networks to mimic the structural equations. Most of the time in causality we seek a Directed Acyclic Graph (DAG) that is consistent with our available data and assumptions, on that end Zheng et al. (2018) proposed a continuous approach to learning DAGs efficiently. A followup work by Brouillard et al. (2020) then combined ideas from both previous works to provide an efficient causal DAG learner.

**Explainable and Interpretable AI/ML.** A great body of work within deep learning has provided visual means for explanations of how a neural model came up with its decision i.e., importance estimates for a model's prediction are being mapped back to the original input space e.g. raw pixels in the arguably standard use-case of computer vision (Selvaraju et al., 2017; Schulz et al., 2020). Formally defined in (Sundararajan et al.), we simply have that $A_F(\mathbf{x}) = (a_1, \ldots, a_n) \in \mathbb{R}^n$ is an attribution of predictive model $F$ when $a_i$ is the contribution of $x_i$ for prediction $F(\mathbf{x})$ (with $\mathbf{x} = (x_1, \ldots, x_n) \in \mathbb{R}^n$). Recently, Stammer et al. (2021) argued that such explanations are insufficient for any task that requires symbolic-level knowledge while comparing the existing state of explanations to "children that are only able to point fingers but lack articulation". (Stammer et al., 2021) therefore proposed a neuro-symbolic explanation scheme to revise and ultimately circumvent "Clever Hans" like behavior from learned models in a XIL user-model loop (Teso & Kersting, 2019). On the causal end, (Schwab & Karlen, 2019) proposed a model-agnostic approach that can generate explanations following the idea of Granger causality (which is very different from Pearlian causality as it captures "temporal relatedness" which holds in their setting as input precedes output). Specifically, they train a surrogate model to capture to what degree certain inputs cause outputs in the model to be explained. They achieve this by simply comparing the prediction loss of the model for the original input $\mathcal{L}(y, \hat{y}_X)$ (where $X$ is the input) with the alternate prediction loss when a certain feature $i$ is being removed $\mathcal{L}(y, \hat{y}_{X \setminus \{i\}})$. On the Pearlian side of explanations, the arguably closest works on explainable AI/ML can be found in research around fairness (Kusner et al., 2017; Plecko & Bareinboim, 2022). For instance, Karimi et al. (2020) investigated how to best find a counterfactual that flips a decision of interest e.g. an applicant for a credit is rejected and the question is now which counterfactual setting (changes to the applicant) would have resulted in a credit approval. From a purely causal viewpoint, our work might be compared to the definitions of Halpern (2016) for "actual causation."

## 3   Deriving a Causal Explanation for the Causal Hans[1] Example

**Causal Hans Example.** Consider the setting from Fig.1 which highlights the Causal XIL loop that we envision. The users, for instance a medical doctor (he) and a ML developer (she), have access to some data sets, for instance a set of patients's medical records. The ML developer will train a model that tries its best in learning about the data distribution's underlying causal relations. The M.D. on the other hand might make an observation such as for the patient named Hans that his mobility is below average, thus raising the question "Why is Hans's Mobility bad?". Given a causal explanation method, we could now derive an explanation that is grounded in the learned causal model to answer the question about the patient Hans in a causal manner—thus the naming Causal Hans example. Not satisfied with the answer, the M.D. decides to tell the ML developer about the inconsistencies he observed between the generated model explanation and his structural intuition. Subsequently, the ML developer adapts the training procedure with the corrections, ultimately resulting in a new learned structure whose generated explanation satisfies the M.D. and moves the learned model closer to the latent true SCM. This completes the Causal XIL setup as we envision, however, what is the causal

---

[1]As analogue and wordplay to the "Clever Hans" notion, see Introduction.

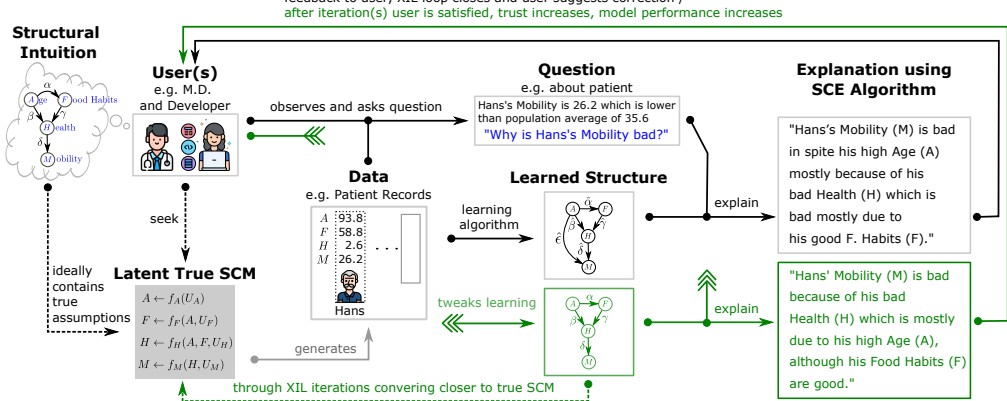

Figure 1: **Causal XIL illustrated on the Causal Hans Example.** Refer to the first paragraph "Causal Hans Example" of Section 3 for a detailed description. We propose the SCE algorithm that allows for truly causal explanations in Causal XIL. (Best viewed in color.)

explanation method? In the following we will derive from first principles a new explanation scheme to answer an observation/question (using our running example about a patient named Hans) that is consistent with a given SCM thereby constituting a causal explanation of a SCM by construction.

### 3.1 DERIVING A COMPUTABLE ALGORITHM FOR CAUSAL EXPLANATIONS

We will use the Causal Hans example that we just introduced as our workbench to derive a computable algorithm for truly causal explanations. For the sake of simplicity we assume that the structural intuition that the user/expert can capture about the problem using their mental model can be described as an SCM[2]. In the following, consider an SCM that generates patients's medical records described by numerical representations for age, nutrition, overall health and mobility respectively ($\mathbf{V} = \{A, F, H, M\}$). Next, let's consider some samples from said SCM. E.g. we might observe the data set containing the individual named Hans $\mathrm{H} = (a_H, f_H, h_H, m_H) = (93.8, 58.8, 2.6, 26.2)$ where each value could be associated with a discrete label e.g. $a_H = 93.8$ would be 93 years old, whereas $m_H = 26.2$ refers to a rather immobile person. The latter label is actually implicitly the assessment $m_H < \mu^M$ where $\mu_M = 35.6$ is the average mobility value for the population, that is, we observe Hans to be a rather immobile person *when compared (or relative) to the whole population.* With this we are in the position to pose a question like

**Q1:** *"Why is Hans's Mobility bad?"*

where the word "bad" refers to "bad relative to the population." Formally, we can now define such an observation or question as:

**Definition 1** (Why Question)**.** *Let $x \in Val(X)$ be an instance of $X \in \mathbf{V}$ of SCM $\mathcal{M}$. Further, let $\mu^X$ be the empirical mean for a set of samples ($\mu^X := \frac{1}{n} \sum_i^n x_i$) and let $R \in \{<, >\}$ be a binary ordering relation. We call $Q_X := R(x, \mu^X)$ a (single) why question if $Q_X$ is true.*

Checking back with the definition, we see that **Q1** defines a valid question for the Causal Hans example since $Q_M := m_H < \mu^M = 26.2 < 35.6$ evaluates to true.

Next, we will discuss the structural intuition of the user (e.g. the M.D. in Fig.1). Generally, the true SCM $\mathcal{M}^*$ is latent but we can realistically expect to have access to partial knowledge (or estimate) of $\mathcal{M}^*$. Say, the user has intuition for an SCM $\mathcal{M}$ that contains the relations $A \xrightarrow{\alpha} F$, $A \xrightarrow{\beta} H$, $F \xrightarrow{\gamma} H$, $H \xrightarrow{\delta} M$ where $\alpha, \beta, \gamma, \delta$ denotes the respective causal effects[3] Further, $\alpha, \gamma, \delta > 0$

---

[2]While several works in cognitive science support the fact that humans are capable of judging causal relationships both qualitatively and quanitatively it is not generally true that the human mental model *is* in fact an SCM but rather that parts of it can be represented by it.

[3]Let's assume that $\mathcal{M}$ is linear for the moment. The key property needed here is just some kind of value estimate for the causal effect, which for linear SCM is simply the coefficients, that is $\alpha_{F \to H} = \gamma$ etc. Thus w.l.o.g. we use it here to make the example more clear.

while $\beta < 0$ meaning that for instance an aging (higher value for $A$) will have a negative, decreasing effect on health (smaller value for $H$), and also $\beta > \gamma$ meaning the causal effect of aging on health is greater in absolute terms than the one of food habits onto health. Now when we intend on answering **Q1** it seems reasonable to start with the queried variable first, mobility in this case. We observe that $M$ is an effect of $H$ with $\gamma > 0$ meaning that since Hans has also below average health (and not just below average mobility) and lower health translates to lower mobility that $m_H$ is inline with $h_H$. Traversing the chain further to the causes of $H$, which are $A$, $F$ in $\mathcal{M}$ we observe two different scenarios. Since $A$ is above average as Hans is an elderly person and $\beta < 0$ we can conclude that $a_H$ is definitely an explanation for $h_H$ whereas $F$ with $\gamma > 0$ is actually a countering factor since Hans has a good diet beneficial to health. In summary, by exploiting the knowledge on $\mathcal{M}$ we have arrived at a causal explanation that can be pronounced in natural language as:

**Explanation 1** (for **Q1**). *"Hans's Mobility is bad because of his bad Health which is mostly due to his high Age although his Food Habits are good."*

Explanation 1 is a truly causal answer to the observation about Hans's mobility deficiency based on SCM $\mathcal{M}$. It captures both the existence and the "strength" of a causal relation. In the following we will capture and formalize our intuition that allowed us to derive Exp.1. This will allows us to move towards computing such causal explanations automatically.

We mainly used four ideas or pieces of knowledge in our argument above: (I) that there is a relative notion in the why question $Q_M$ like "why … bad?" that implicitly compares an individual (here, Hans) to the remaining population, (II) note that by definition there can only exist a causal effect from some variable to another *if and only if* one is the argument of the other in a structural equation of $\mathcal{M}$, (III) the causal effect $\alpha_{X \to Y}$ allows us to assert whether the observed values for $(x, y)$ are "surprising" or not (e.g. it was not surprising that $m_H < \mu^M$ after observing $h_H > 0$ and knowing that $\gamma > 0$ since decreasing health means decreasing mobility in general and Hans is old), and (IV) that some causal effects are more important or influental than others (e.g. age versus food habits w.r.t. health). We can neatly collect all information from (I-III) in a single tuple which we call causal scenario.

**Definition 2.** *The tuple $C_{XY} := (\alpha_{X \to Y}, x, y, \mu^X, \mu^Y)$ is called causal scenario.*

The (IV) point we can capture separately as will be shown below. Now, we finally express our build up intuition and understanding into rules expressed in first-order logic that will then allow us to compute causal explanations like Exp.1 automatically.

**Definition 3** (Explanation Rules). *Let $C_{XY}$ denote a causal scenario, let $s(x) \in \{-1, 1\}$ be the sign of a scalar, let $R_i \in \{<, >\}$ be a binary ordering relation and let $\mathcal{Z}_X = \{|\alpha_{Z \to X}| : Z \in \mathrm{Pa}_X\}$ be the set of absolute parental causal effects onto $X$. We define FOL-based rule functions as*

*(ER1)* *If $R_1 \neq R_2$, then:* $\quad R_1(s(\alpha_{X \to Y}), 0) \wedge (R_2(y, \mu^Y) \vee R_1(x, \mu^X))$,

*(ER2)* *If $R_1 \neq R_2$, then:* $\quad R_1(s(\alpha_{X \to Y}), 0) \wedge R_1(y, \mu^Y) \wedge R_2(x, \mu^X)$, *and*

*(ER3)* *If $|\mathcal{Z}_X| > 1$, then* $\quad Y \iff \arg\max_{Z \in \mathcal{Z}_X} Z$

*indicating for each rule $ERi(\cdot) \in \{-1, 0, 1\}$ how the causal relation $X \to Y$ satisfies that rule.*

One can easily verify that these rules form the building blocks for the generation of causal explanation like in Exp.1. We can now give a simple recursive algorithm that traverses all possible directed paths (or causal chains) to the queried variable checking each of rules *ERi* thus constructing a unique code that maps to a unique answer. We define the Structural Causal Explanation algorithm as:

**Definition 4** (SCE). *Like before let $Q_X$, $\mathcal{M}$ be a valid why-question and some proxy SCM. Further, let $\boldsymbol{D} \in \mathbb{R}^{n \times |\mathbf{V}|}$ denote our data set. We define a recursion*

$$\mathbf{E}(Q_X, \mathcal{M}, \boldsymbol{D}) = (\bigoplus_{Z \in \mathrm{Pa}(X)} ER(Z \to X), \bigoplus_{Z \in \mathrm{Pa}(X)} \mathbf{E}(Q_Z, \mathcal{M}, \boldsymbol{D})) \tag{1}$$

*where $\bigoplus_{i=1}^{n} v_i = (v_1, \ldots, v_n)$ denotes concatenation and ER checks each rule ERi (Def.3), and the recursion's base case is being evaluated at the roots of the causal path to $X$, that is, for some $Z \in \mathbf{V}$ with a path $Z \to \cdots \to X$ we have*

$$\mathbf{E}(Q_Z, \mathcal{M}, \boldsymbol{D}) = \emptyset. \tag{2}$$

*We call $\mathbf{E}(Q_X, \mathcal{M}, \boldsymbol{D})$ Structural Causal Explanation of $\mathcal{M}$.*

**Causal Hans Example revisited (using SCE algorithm).** To return one last time, we clearly see that for $Q_M$ (corresponding to **Q1**) we can compute using Eq.1

$$
\begin{aligned}
\mathbf{E}(Q_M, \mathcal{M}, \boldsymbol{D}) &= ((ER1 = -1), \bigoplus_{Z \in \{A,F\}} \mathbf{E}(Q_H, \mathcal{M}, \boldsymbol{D})) \\
&= (\ldots, (((ER1 = 1, ER3 = 1), \mathbf{E}(Q_A, \mathcal{M}, \boldsymbol{D})), ((ER2 = 1), \mathbf{E}(Q_F, \mathcal{M}, \boldsymbol{D})))), \\
&= (\ldots, ((\ldots, \emptyset), (\ldots, \emptyset))).
\end{aligned}
$$

So the recursion result is $H \to M : (ER1 = -1, ER2 = 0, ER3 = 0), A \to H : (ER1 = 1, ER2 = 0, ER3 = 1), F \to H : (ER1 = 0, ER2 = 1, ER3 = 0)$. This result *uniquely* identifies the human understandable pronunciation of our causal explanation in Exp.1. We provide a detailed explanation on the pronunciation scheme and also intuitive namings for the rules $ER_i$ in the appendix. It is worthwhile noting that the natural language choice of words to express the interpretation is not implied by the form of the SCE e.g., while Hans's mobility is said to be "bad", a car's remaining fuel is rather considered to be "low".

## 3.2 THEORETICAL PROPERTIES OF *ER* AND SCE

The concepts of why-question, causal scenarios and *ERi* rulest hat we had to develop for the introduction of SCE algorithm, alongside SCE itself, come with several mathematical consequences which we now discuss. All of the subsequent results are simple and can be proven easily, still, their importance needs to be stressed since they make implications about the wide applicability of SCE.

**Proposition 1.** *For any causal scenario the rules ER1 and ER2 will be mutually exclusive.*

*Proof.* First, we code the binary ordering relations $<, >$ to represent 0 and 1 respectively. Second, we observe that $ERi \in \{<, >\}, i \in \{1, 2\}$ always involves the triplet $T = (R(s(\alpha_{X \to Y}), 0), R(y, \mu^Y), R(x, \mu^X))$. Third, let $\mathbb{T} := \{0, 1\}^3$ be the set of all such triples as their code words, so $T \in \mathbb{T}$. Looking at the total number of possible scenarios $|\mathbb{T}| = 2^3 = 8$, we easily see that ER1 covers codewords $\{010, 011, 100, 101, 000, 111\}$ and ER2 covers the codewords $\{001, 110\}$, and together they cover all codewords $ER1 \cup ER2 = \mathbb{T}$. Since any single scenario $C_{XY}$ is uniquely mapped to a codeword, it will either trigger ER1 or ER2 but never both. $\square$

**Proposition 2.** *The SCE recursion always terminates.*

*Proof.* The recursion's base case is reached when a root node is reached i.e., a node $i$ with $\mathrm{Pa}_i = \emptyset$. An SCM implies a finite DAG, so root nodes are reached eventually. $\square$

**Theorem 1.** *The output of any causal structure learning algorithm can be used to compute SCE.*

*Proof.* The proof for this theorem is surprisingly simple in that the SCM $\mathcal{M}$ used in the SCE recursion is only required to provide some kind of numerical value $\alpha_{i \to j}$ for the relation of any variable pair $(i, j)$, that is, a matrix $A \in \mathbb{R}^{|\mathbf{V}| \times |\mathbf{V}|}$ which represents a linear SCM or a SCM where each $\alpha_{i \to j}$ represents a causal effect description. If the matrix $A$ is an adjacency matrix living in $[0, 1]^{|\mathbf{V}| \times |\mathbf{V}|}$, then we simply have no information about ER3 since all causal effects are assumed to be the same. Since any causal structure learning algoirthm will produce a causal graph represented by a matrix, we have that we can compute SCE. $\square$

The beauty of Thm.1 can be fully appreciated when being put into the context of practical AI/ML research and application. It tells us that *any* causal graph learner ever invented and that will ever be invented can provide causal explanations on any query of interest consistent with the learned model thus reflecting the learnt. In practice this means that all prominent graph learning algorithms like NT (Zheng et al., 2018), CGNN (Goudet et al., 2018), DAG-GNN (Yu et al., 2019) and NCM (Ke et al., 2019) are all explainable[4]. On a concluding note to this section, we have a remark on SCM that allow for hidden confounder. SCE as presented Def.4 do not cover hidden confounders and we leave this for future work. However, we can always modify the algorithm to talk about "unknown reasons" when giving knowledge on $\mathbf{U}$. An extended discussion on this and also other noteworthy aspects of SCE can be found in the Appendix.

---

[4]The DAG learner in NT can be interpreted as a linear SCM but there is no guarantee.

Figure 2: **Comparison of SCE with CXPlain.** Refer to the text in Sec.4 for the discussion. Both approaches use the same ground truth SCM, however, only SCE is capable of leveraging necessary causal knowledge. (Best viewed in color.)

## 4 EMPIRICAL ANALYSIS

### 4.1 FAILURE OF EXISTING EXPLANATION METHODS

In the beginning we said that existing methods such as CXPlain (Schwab & Karlen, 2019) would fail to provide truly causal explanations useful for Causal XIL but did not argue or show *how* they would fail. Now after having introduced our solution to the problem in the previous section on SCE, we can compare both and make apparent the shortcomings that we have fixed with SCE. For illustration we make once again use of the Causal Hans example. Fig.2 shows the results. We provided CXPlain with the same ground truth SCM as SCE to train the surrogate explanation model. We trained 10 bootstraped neural models using suitable parameters for the masking operation and loss function. What we observe is a distribution over importance scores where all factors are being deemed relevant and "causal" to the output (which in this case is the mobility of Hans). Also the highest attribution is given to age, then food habits and finally the lowest to health. This single observation makes apparent two important shortcomings: **(I)** from the output we do not know which is a direct $(H)$ and which are indirect $(A, F)$ causes while the ordering of presentation in SCE clearly distincts the former from the latter, and **(II)** we have no information on the causal effect, that is, we cannot tell in which way a variable with high attribution will affect the predicted variable, for example food habits received a high importance score like age but age will have a detrimental effect on mobility whereas food habits will have a beneficial effect—again, SCE fixes this by discriminating the positive and negative cases. We also ran CXPlain for more queries and observed two further fallacies that we discuss in the Appendix. One the flip side, like SCE, the CXPlain attribution was able to identify the that the effect of aging is stronger that that of dieting.

### 4.2 QUALITY OF LEARNED EXPLANATIONS

The overarching question for this experiment was "If we apply a causal graph learning method to our data, then what do the generated explanations reveal about the learning method?" To investigate this question we looked at different data sets including different graph learners and for each combination their respective SCE. We considered several different why-questions for each of the four data sets: data set for the Causal Hans example, weather forecast (W, real world, Mooij et al. (2016)), mileage (M, synthetic), and recovery (R, real world, Charig et al. (1986)). To avoid cluttering in the main text we have moved the relevant tables and figures to the the Appendix where we also provide an extended account. Here we highlight what the SCE when applied to NT (Zheng et al., 2020) would reveal about what was learned. We ran NT with suitable paramaters. In Tab.2 of the Appendix we see the computed SCE. We observe expected results on the W and M data sets, whereas differences on the R and H data sets. For R, the difference is only subtle as the model's explanation to the query "Why did Kurt not Recover?" is not "Kurt did not Recover because of his bad Pre-condition, although he got Treatment." but "[...], which were bad although he got Treatment." which is on the second recursion in the reasoning process i.e., the treatment countering the state of condition and not affecting the condition itself. This difference becomes apparent in the graphical structure where the arrow from Pre-conditions to Treatment is inverted contrary to expectation. To illustrate one more example using the data set to our Causal Hans example where the difference was more drastic, here the discrepancy revolves around a totally different graph structure e.g. the learned model expects a direct cause-effect relation between age and mobility while also wrongly assuming that food habits have a detrimental effect on health. Therefore the answer to the question "Why is Hans's Mobility bad?" suddenly becomes "Hans's Mobility, in spite his high Age, is bad mostly because of

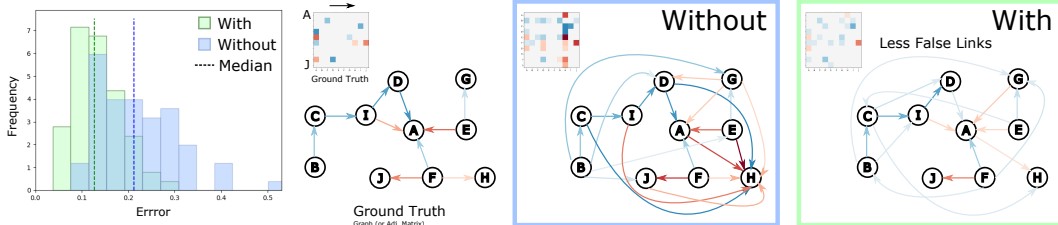

Figure 3: **Graph Learning Improves with Explanations.** Left: error distributions when performing graph learning with/-out SCE regularization (which is simply an added penalty term for inconsistent explanations), next to is the ground truth graph. Right (boxes): the predicted graphs, showing a decreased number of false positives. (Best viewed in color.)

his bad Health which is bad mostly due to his good Food Habits." which sounds very absurd. The ground truth SCM for this data set contains non-linear causal relationships, while NT makes linearity assumptions, which explains the wrongly learned graph structure. The crucial observation, though, which answers our initial question about what SCE can reveal about the learning algorithm is the following: only by looking at the SCE, effectively using it as a graph distance or metric, we were able to tell that the learned model is very different from what we expected. Put differently, it made apparent for the Causal Hans example that by simply adding an extra edge (here between $A$, $M$, see Tab.2) and flipping the sign of an edge (here between $F$, $H$) we already get a big difference in what these graphs express/explain.

### 4.3 USING EXPLANATIONS AS REGULARIZATION DURING LEARNING

Since SCE contain knowledge on (some) causal relationships underlying the data, we wondered to which extent they might be used for improving the sample efficiency of graph learners. In this experiment the overarching question therefore was "If we have explanations available provided, by say an expert, then could we use it for improving learning?" For ease of applicability and consistency with the previous experiment we take NT again and add a simple regularization term to its loss that penalizes inconsistent explanations. We generate 70 random linear SCMs with respective observation distributions. Then we use graph learning to infer 70 more graphs, making 140 graphs in total. For each graph we generate 50 random single-why questions to be answered, resulting in a data set of 7000 explanations. All the details regarding this learning setup, such as for instance how to make make SCE differentiable for it to function as training signal, are being discussed in the Appendix. The graph learning is being performed in a data scarce setting with only 10 data samples per graph. Thus to infer the true causal structure the method ideally needs to perform sample-efficient learning. Fig.3 shows our results. The error distributions over all of the graphs are shown both with and without the SCE regularization. We also highlight the graph estimate upon which most improvement was observed. It can be observed that with the regularization the method can both identify more key structures while significantly reducing the number of false positives. For example many false links that pointed towards node H (like B to H or G to H) were removed while some key structures could now be recovered like the directed edge from node I to node A. While more experiments would be necessary to claim that indeed learning is (significantly) improved through explanations, our naïve learner already provides evidence in favor of that initial hypothesis that sample efficiency is being improved by models that are under pressure to explain what they learned.

### 4.4 SURVEY WITH 22 PARTICIPANTS

Throughout this paper we provided several arguments in advocacy of Causal XIL as the key paradigm of interest for future research and application. We proposed SCE as a solution to the problem of truly causal explanations, therefore, in this final experiment we investigate the overarching question of "What does SCE explain about the causal intuition that humans have that could provide for Causal XIL?" We let $N = 22$ human subjects judge the qualitative causal structure of four "daily-life" examples using a questionnaire specifically designed to provide us with the data necessary for constructing causal graphs representative of what the participants think about the presented concepts. Please refer to the Appendix for the questionnaire and [Human Data] for the anonymized

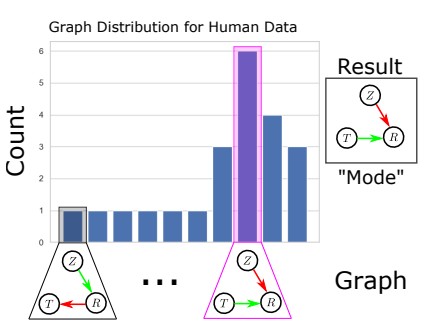 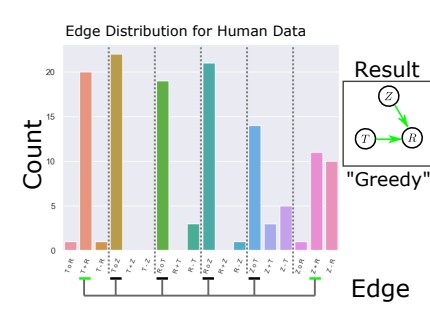

Figure 4: **Measuring Agreement between Subjects.** Refer to Sec.4.4 for details. Left, the graph estimate is the mode of the distribution. Right, greedily pick each edge. (Best viewed in color.)

answers that we used for evaluating the survey. Also a prolonged discussion is provided in the Appendix. The first question to answer is, how did we construct the graph estimates from human data? In Fig.4 we show two ways that we considered: "Mode" where we can simply look at the different graphs and take the most frequently occurring one as representative of the population or "Greedy" where we look at the frequency at which edges are predicted and then simply construct a graph from greedily taking the most probable edge each time. Greedy comes at the cost that the predicted graph is not necessarily in the populaton. With the graphs at hand, we can analyze our initial question. For brevity, we will only highlight some key observations: **(I)** we observe a systematic approach and thereby non-random approach to edge-/structure-selection by the subjects. Furthermore, there are only a few clusters even with increasing hypothesis space. Both the systematic manner and the tendency to common ground are evidence in support of SCMs and mental models overlapping, **(II)** we observe that the increase in hypothesis/search space (i.e., more variables) comes with an increase in variance. This variance increase can be argued to be due to the progressive difficulty of inference problems as well as decreased levels of attention and potential fatigue across the duration of the experiment, **(III)** some subjects implicitly assume a notion of time for example that there must be a cyclic relationship between, say, treatment and recovery where the subject likely thought in terms of 'increasing treatment increases the speed of recovery *which subsequently* feeds back into a decrease of treatment (since the individual is better off than before)', and **(IV)** answering our initial question of looking at the SCE we can observe that they lie much closer to the expected explanation opposed to what we have observed in our second experiment when looking at existing methods for graph learning (Sec.4.2), suggesting that the human subjects are better at judging causal relationships for the examples we considered.

## 5 CONCLUDING REMARKS

Our work made clear how existing explanation methods, even when proclaimed causal, leave open the problem of truly causal explanations. We further argued that truly causal explanations are need of the hour since we ultimately seek a causal variant to XIL as it can improve both model performance out-of-distribution and increase user trust. To this end, we derived from first principles using our Causal Hans example a computable explanation algorithm grounded in SCM. We proved the wide applicability of our SCE algorithm and then went on to corroborate our results with several experiments including a naïve approach to learning with explanations and a survey with 22 participants to analye the human component in XIL. For future work, there are several interesting routes to take such as proving that the proposed ER$i$ rules are complete in that there is no other missing rules in the formalism or also reasoning about "when you don't know" to capture for example uncertainty about the inputs like the current belief about the SCM.

**Societal / Ethical Implications.** While this work discussed for the first time a XIL approach with truly causal explanations and therefore is to be considered still in its infancy, deploying a causal XIL process with SCE could potentially help in many applications to improve model performance and increase user trust by making transparent the "understanding" of the model in natural language.

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

# A APPENDIX FOR "CAUSAL EXPLANATIONS OF STRUCTURAL CAUSAL MODELS"

We make use of this appendix following the main paper to provide extended discussions and details.

## A.1 LEARNING DAGS AND CAUSAL GRAPHS

Induction of inter-variable relationships based on available data lies at the core of most scientific endeavour (Penn & Povinelli, 2007). The sub-class of relation structures of DAGs plays a central role due to its representational role in causality (Pearl, 2009; Peters et al., 2017). Unfortunately, due to the combinatoric nature of the problem setting, learning DAGs from data is recognized to be an NP-hard problem (Chickering et al., 2004). In their seminal work, Zheng et al. (2018) were able to re-formulate the traditional view into a continuous shape such that any non-convex optimization can be applied for the graph estimation problem. The authors propose the general formulation, $\min_{\boldsymbol{W} \in \mathbb{R}^{d \times d}} f(\boldsymbol{W})$ subject to $h(\boldsymbol{W}) = 0$, where $f$ is a data-based score, e.g. in Zheng et al. (2018) a regularized least-squares loss is applied assuming a sparse linear model (possibly SCM) i.e., $f(\boldsymbol{W}) = ||\boldsymbol{X} - \boldsymbol{X}\boldsymbol{W}||_F^2 + ||\boldsymbol{W}||_1$, and $h$ is a smooth function with a kernel (or null space) that only contains acyclic graphs, $h(\boldsymbol{W}) = 0 \iff \boldsymbol{W}$ is acyclic. Different variations of the same continuous counting mechanism using this acyclicity constraint have been proposed, e.g., Zheng et al. (2020) proposed $h(\boldsymbol{W}) = \mathrm{tr}(e^{\boldsymbol{W} \circ \boldsymbol{W}}) - d$ while Yu et al. (2019) proposed $h(\boldsymbol{W}) = \mathrm{tr}[(\boldsymbol{I} + \boldsymbol{W} \circ \boldsymbol{W})^m] - m$, unfortunately, both suffer from cubic runtime-scalability in the number of graph nodes, $O(d^3)$. While the aforementioned works have focussed on data originating from (non-linear transformation) of linear SCM, there exists yet another sub-class of DAG-learning methodologies that focuses on more general causal inference. Ke et al. (2019) made use of data from the first two levels of Pearl's Causal Hierarchy (PCH; Pearl (2009); **?**), namely observational and interventional, to update their graph estimate while using masked neural networks to mimic the structural equations. Brouillard et al. (2020) follows the same idea of leveraging causal information, e.g. interventional data, for overcoming identifiability issues while staying close to the continuous optimization formalism introduced in (Zheng et al., 2018).

## A.2 FAIL OF EXISTING METHODS, EXAMPLE SHOWCASE WITH CXPLAIN

In the main text we illustrated the key aspects of how (even proclaimed causal) explanation methods like CXPlain (Schwab & Karlen, 2019) fail to do justice for a causal XIL framework. Here we highlight two more interesting shortcomings. Fig.5 shows the same setup as from the main text but for different queries (i.e., patients other than Hans). We make the following observations: (I) the attributions are deterministic. This might first be considered a feature, however, the causal mechanism of an SCM are only deterministic up to a realization of the exogeneous terms. Therefore, we can have the exact same patient record for *different patients*. This cannot be captured by these previous attribution methods. (II) when querying for random individuals we actually observe inconsistencies between the attributions which is weird since the patient records are being generated by the same causal mechanisms. For example in the Hans case we had age, nutrition and then health ordered from highest to lowest attribution. For a rather similar patient we observe that nutrition and age swap in importance. For yet another patient we observe that suddenly age and nutrition, which previously played the most important role, are not important anymore. The shortcomings (I,II) corroborate the shortcomings previously mentioned in the main text, rendering the need for truly causal explanations as presented with SCE more so important.

## A.3 HYPOTHESIS: MENTAL MODEL OF HUMANS REPRESENTABLE AS SCM?

It has been argued that at the core of a human mental model (abbreviated MM in the following) the illustration of one's thought process (regarding the understanding of world dynamics) is to be found (Simon, 1961; Nersessian, 1992; Chakraborti et al., 2017). The difficulty of said thought process illustration is partly due to circular and abstract terms like explanation and interpretations for which we do not provide an explicit definition as this is up to philosophical debates and ideally we keep the idea more general than what has been done previously in explainable AI/ML where "explanation equals pixel attributions" in many cases. Assuming the world dynamics to be governed by causality we observe that humans are capable of modelling both causal relationships between endogenous

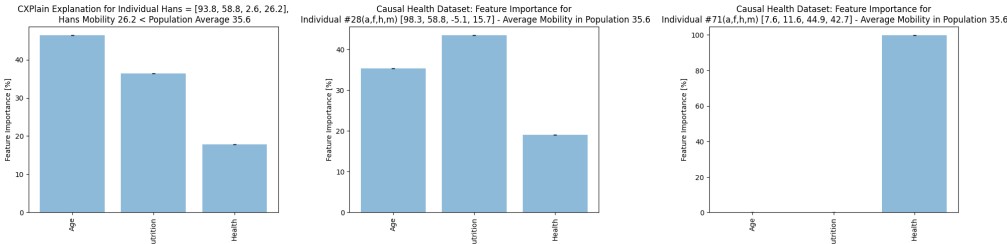

Figure 5: **Further Examples of CXPlain Shortcomings.** Refer to Sec.A.2 for details.

variables and additionally information on the strength of said relationship. Put differently, MM model a causal graph and corresponding causal effects akin to the formal notions from the previous section. Consider the following real world example:

> **MM Example.** *At any given time a human has a state of overall health (relating to fat-muscle ratio, allergies and diseases, etc.) and mobility (relating to the general freedom and flexibility of movement, e.g., a gymnast is more mobile than the average person). Now, the MM allows inferring (1) that mobility is being (partially) caused by something else (for instance health, e.g., being overweight decreases one's mobility) and (2) that different events can have different "strength" e.g., that an average car accident causes more harm to the individual's mobility than an average workout session causes good.*

A natural candidate for capturing the two properties from the MM example formally are SCM, thereby we hypothesize the following:

**Hypothesis 1** (**MM Conversion, short MMC**). *The parts of the MM that are being used for encoding the causal relationships of the variables of interest can be formally captured by a corresponding SCM, in short this "equivalence" can be denoted as MM ≡ SCM.*

While the MMC hypothesis leaves room for notions not captured by mathematical rigor, it suggests an equivalence to SCM regarding the causal aspects. The MM example has motivated the MMC hypothesis which itself suggests *a justification of using SCM in the first place*.

## A.4 IMPLICATIONS OF MM ≡ SCM

If we accept that MM ≡ SCM, then we can use SCMs as an adequate proxy to the MM. Furthermore, any useful property of SCM implies corresponding aspects back in the MM. We immediately observe one such key property of SCM namely comparability. That is, if one is given say two different SCMs that are defined over the same endogenous and exogenous variables (so only differing in the actual parameterizations) then one can compare said SCMs i.e., there exists a notion of distance. For the linear case, we can easily prove this by constructing an example metric space.

**Definition 5.** *We define a function $d(\mathcal{M}_1, \mathcal{M}_2) = \sum_{i \neq j} |\mathcal{M}_1(i,j) - \mathcal{M}_2(i,j)| + q(P_1, P_2)$ where $q$ is the square-root of the Jensen-Shannon Divergence (JSD), $\mathcal{M}_k = \langle \mathbf{U}_i, \mathbf{V}_i, \mathcal{F}_i, P_i(\mathbf{U}_i) \rangle$ for $k \in \{1, 2\}$ such that $\mathbf{V}_1 = \mathbf{V}_2$, $\mathbf{U}_1 = \mathbf{U}_2$, $\mathcal{F}_k$ define linear functions in $\mathbb{R}$, and in slight abuse of notation $\mathcal{M}_k(i,j)$ is the causal effect $\alpha_{i \to j}$.*

**Proposition 3.** *Let $d$ be as in Def.5 and let $\mathbb{M}$ denote the set of all linear SCM defined over the same exogenous and endogenous variables, $\mathbf{U}, \mathbf{V}$. Then $(\mathbb{M}, d)$ is a metric space.*

*Proof.* The absolute difference on the real numbers is a metric (i.e., positive-definiteness, symmetry, and triangle-inequality hold) therefore holding for the "dependency" terms from $\mathcal{F}$. Furthermore, $q$ was chosen as the Jensen-Shannon-Metric. Finally, metrics are closed under summation. □

Prop.3 is just one example of what might be considered a sensible metric space for a subset of all SCMs. What it does is compare each of the linear coefficients for any causally related tuple of variables, aggregating the sum, and further adding a divergence term between the defined distributions

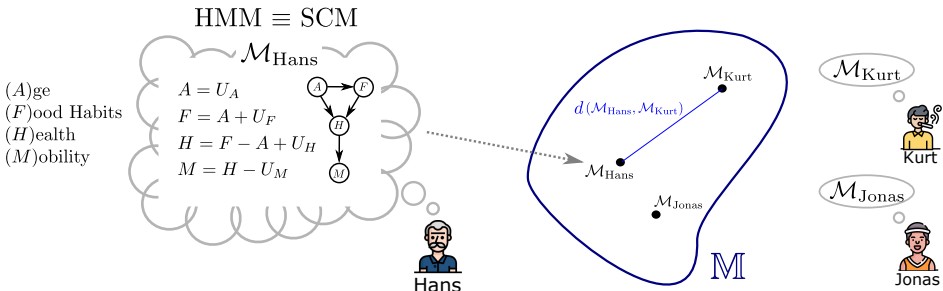

Figure 6: **MMC Hypothesis and Linear SCM Metric Space.** Left: Accepting Hyp.1 means that the MM of Hans is an SCM. Right: Different linear SCM (from different individuals) can be compared, an example metric space for $(\mathbb{M}, d)$ is given by Prop.3. (Best viewed in color.)

over the exogenous variables. This comparability and the visual intuition behind MMC are illustrated in Fig.6. We now state our *first key observation* following Hyp.1 and Prop.3: the existence of a "true" SCM is in fact justified i.e., there *exists* an underlying data generating process for any data and the MM of any person might or might not coincide with that SCM.

On another note, consider the fact that while the "true" SCM represents the concept of objectiveness, oppositely, the MMs are of subjective nature (that is, every human has their own subjective life experience). Coming back to MM ≡ SCM, we see that Prop.3 further implies that MMs are also capable of dis-/agreeing with each other. With this at hand, we now state our *second key observation*: in most practical cases having access to many SCM-encodings of subjective MMs can ultimately lead in their overlap-agreement to (parts of) the objective "true" SCM. There is certainly no guarantee since all available MM-SCM samples can in fact be wrong, however, note the emphasis on *in most practical cases*—therefore, identifying this overlap in MM (or SCM) for a specific problem is highly valuable for AI/ML research.

Our final, *third key observation* is concerned with explanations. Existing literature views explanations as *derivable* from MMs and thus implicitly containing some information on the MM (Chakraborti et al., 2017) and since MM ≡ SCM, we argue that there must exist an equivalent of the human notion of explanation *within SCM*. This justifies our further investigation on SCM-based explanations, which eventually leads to the formalism of SCE. The benefits of an approach using explanations derived from SCM are two-fold (1) that by construction they are human understandable allowing for explainable ML in which models can reason about the learnt and (2) that the models themselves become better, as they need to account for consistency in explanations, which is beneficial to any downstream-task.

## A.5    ELABORATION ON SCE PROPERTIES

The three basic logic rules (Prop.1) dictate how the SCE (4) will look like for some causal estimate of the system and any given query and data. Having the actual relation $R$ as a return argument of each of the rules allows for a fine-grained explanation. In a nutshell, it allows to extend a statement "Y because of X" to a more detailed one like "Y because of X being low". The general pronunciation scheme for the the three rules (excitation, inhibition, and preference), that allows for a human-understandable natural language version of the SCE, are as follows: The pronunciation of the details

| | | |
|---|---|---|
| *ER*1 | Excitation | "Y because of X [being low/high]" |
| *ER*2 | Inhibition | "Y although X [is low/high]" |
| *ER*3 | Preference | "mostly" + *ER*1 or *ER*2 pronunciation |

Table 1: **Pronunciation Scheme.** Right shows the natural language reading of a rule's activation.

to the relation e.g. "low"/"high" is context-dependent in that these words might need to replaced with adequate/corresponding words suitable for the context i.e., "the Matterhorn is cold because of the high altitude" ("cold Temperature because of Altitude being high") is fine while "the remaining car fuel is low because of the bad driving style" ("low Fuel because of Driving Style being bad")

requires the context-adaptation ("low" is converted to "bad"). Another noteworthy detail to the SCE properties is the property of *non-repeating causes within explanations* which reduces redundancy. Consider for instance our lead example on Hans's mobility (Exp.1 or Fig.1), the SCM suggests that $F$ can also be explained by $A$, since $A \to F$. However, the corresponding SCE does not give this reason because of the aforementioned property which ensures that redundancy is being avoided. I.e., in the explanation step before we actually explain $H$ using both $A$ and $F$, since $\{A, F\} \to H$, therefore, making it irrelevant for the question to explain the relation between the parents $(A, F)$. While we provided intuition on the derivation of these basic FOL rules alongside the "Causal Hans" example, we now additionally motivate the namings "excitation", "inhibition" and "preference" i.e., what inspired us to name them in such way. We took inspiration from *neuroscience*, where the former two terms relate to the way neurons interface with each other using their synaptic-dendric connections. The last term is a term to propose "relativity" and thus a preference for one cause over the other. All terms thereby adequately describe any causal path in qualitative terms while also providing an almost synonym-quality to the pronunciations (Tab.1).

## A.6 DETAILS FOR EXPERIMENT 2

We select NT (Zheng et al., 2018) as a representative data-driven graph learner for the illustration in Tab.2 which considers the data sets and why questions illustrated in the appendix Fig.**??** i.e., weather forecast (W, Mooij et al. (2016)), health (H), mileage (M), and recovery (R, Charig et al. (1986)). The SCE generated using the learned causal semantics are identical for the DW and M data sets, while differing only subtle for R and drastically for CH data sets. The former discrepancy occurs on the second-level of reasoning i.e., the right top-level explaining answer is given to the question (i.e., "Kurt did not recover because of the problematic pre-conditions") but was contrasted wrongly (i.e., the treatment countering the state of condition and not affecting the condition). The latter discrepancy revolves around a totally different structure e.g. the learned model expects a direct cause-effect relation between age and mobility while also wrongly assuming that food habits have a detrimental effect on health. An explanation in the case of NOTEARS is clearly the violation of the linearity assumption for the CH data set generating SCM.

While in Thm.1 we prove that graph learner are generally explainable in the sense of SCE, for empirical illustration we also provide more examples of such graph learner-based SCE, as we did with our lead examples for NT, in this case additionally for CGNN (Goudet et al., 2018) and DAG-GNN (Yu et al., 2019). Tables 2 and 4 shows an application to NT with graph visualizations and of all methods to a superset of questions (that is, same and more) as the data used for NT. It is crucial to note that the presented results have *not* been hyperparameter-optimized (HO). Take for example CGNN, where candidate selection is exhaustive (brute force, and thus super-exponential in the number of nodes) and the model selection heavily relies on the neural approximation, thereby, HO is likely to be important. In a nutshell, the motivation behind Tab.4 is to present support for our theoretical proof on SCE-interpretability of graph learner i.e., we also give empirical proof for several methods in practice (opposed to pure theory). To assess the quality of the SCE, it is important to note the assumptions made by the original method. E.g., NT and DAG-GNN assume linear SCM. Thereby, we have *no guarantees* for running such a method in a non-linear data domain (which we do with the data sets DW and CHD). *Interestingly, these assumptions can in fact be exposed by SCE.* Consider the DW data set (Tab.4, first example), theory suggests that a linear model with Gaussian noise will exist in both directions $X \to Y$ and $Y \to X$, thus being non-identifiable (Peters et al., 2017). Methods like NT and DAG-GNN therefore pose the assumption that the given data comes, in this case, from a linear model with Gaussian noise i.e., the identifiability problem is being circumvented altogether. This is also the reason why different random seeds can lead to both modellings ($A \to T$ and $T \to A$) for the DW data set (see in Tab.4 how the SCE flips for $\mathcal{M}_3$=DAG-GNN for the two opposing DW queries). Another important note is that the uninformed SCEs "No causal explanation ..." occur when the method's SCM estimate does not contain a causal path to the variable that is being queried by the why question i.e., the SCM will actually contain a non-trivial estimate of the underlying causal structure, even though the SCE returns a trivial/empty explanation since the variable of interest can not be reached within the estimate's structure with a directed path (i.e., the base case in Def.4 is trivially triggered). In fact, these negative "no answer"-type of cases are important since the model need also be able to know when there is nothing to be known. For this case, we also pose why questions to which the ground truth is already a "no answer" explanation since there is no causal connection to the variable being queried by the why-question. The empirics

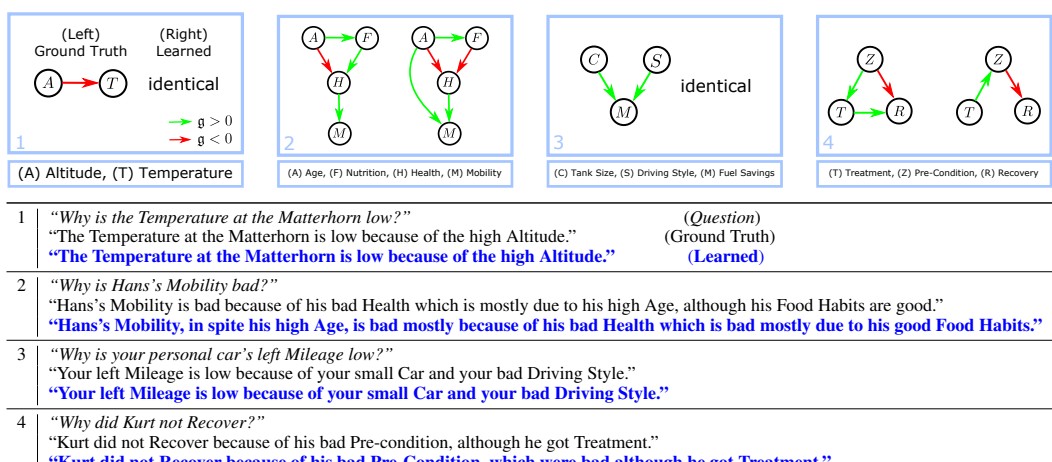

Table 2: **Quality of Learned Interpretations.** We chose the simple, popular NT from Zheng et al. (2018) as our graph learner for generating the SCE. Subtle differences between explanations exist e.g., the explanation 4 is right on the top-level but for the wrong reasons, that is $T \rightarrow Z$ instead of $T \rightarrow R$. Variable letters are capitalized. (Best viewed in color.)

in Tab.4 suggest, as theoretically proven (Thm.1), that the graph learner are explainable and also that all 3 rules (excitation, inhibition and preference) are being used for the graph learner-based SCE. As a positive example, consider example #3 for the CH data set where $\mathcal{M}_1$ captures the complex explanation correctly up to preference and falsely assuming that food habits ($F$) have a negative causal effect on health ($H$). A more interesting example (#8 for the R data set) shows that the main reason being bad pre-conditions ($Z$) is being captured but the model falsely assumes that those are because of the received treatment ($T$). To consider a negative example have a look at example #4 again for CH where the actual answer is a "no causal explanation" since age ($A$) is a root node. However, $\mathcal{M}_3$ claims that the age is high because of the food habits and mobility ($M$), then again because of health. While the statement is wrong and also feels exaggerated, inspecting closely one can detect the correct existence of the causal edge between mobility and health ($H \rightarrow M$). I.e., the model interprets wrongly, but its causal model is still partially valid.

## A.7 Details for Experiment 3

We again make use of the NT as representative of graph learners for the subsequent experiment in which we investigate our conjecture from the final paragraph in Sec.3.2 i.e., whether SCEs themselves can be used as a supervision signal to improve the quality of the learned graph. To circumvent the non-differentiable nature of our recursive formulation of SCE (consider checking the formalism again in Def.4) we train a neural network on a set of legal SCE to mimic the interpreter while being fully differentiable. Following Zheng et al. (2018), we generate 70 random linear SCMs following Erdos–Renyi structures. We use graph induction to infer 70 more graphs, making 140 in total. For each graph we generate 50 random single-why questions to be answered, resulting in a data set of 7000 explanations. We extend the NT loss composition with this neural approximation using a SCE regularization penalty (to penalize SCE inconsistent graph estimates) and perform graph induction once with and once without the regularization (where between 1 and 50 explanations are being observed). The graph induction is being performed in a data-scarce setting with only 10 data samples per graph induction. Thus to infer the true causal structure the method ideally needs to perform sample-efficient. Main paper Fig.3 shows our empirical results on the error distributions for all the graphs while presenting the qualitative difference in the estimated graphs for the most significantly improved example. It can be observed that with the regularization the induction method can both identify more key structures while significantly reducing the number of false links, thereby appearing to be overall more sample-efficient. An explanation would be that, as conjectured, the explanations contain valuable information about the underlying SCM if the explanations themselves were generated by a similar SCM, thereby striking structures that would lead to contradicting explanations.

A.8   DETAILS FOR EXPERIMENT 4

We instructed $N = 22$ participants to answer our questionnaire (see appendix Fig.7). The questionnaire asked the following questions: *"Given a pair of variables, does a causal relationship exist (existence)? If yes, then which is the cause and which is the effect (direction)? If there are multiple causes for a single variable, then how impactful is each of the causes (preference)?"* All of these questions, alongside their responses, are of *qualitative and subjective* nature. It is important to note that the participants *do not* perform the actual induction from specific, provided data like the algorithms do i.e., the human subjects are not given the variable names nor concrete data points that would allow them to find the rules for the specific data sets. Instead, they were only given the variable names/depictions, thereby having to induct from personal experience/understanding essentially. This approach to human induction is related to the experimental setups in (Griffiths & Tenenbaum, 2006; Hattori, 2016).

The motivating lead research questions we intended to answer, and in fact do answer successfully with this experiment, are: What are SCM that (some) human could model? How does overlap for human-based SCM occur? How do subsequent SCE (Def.4) between humans and algorithms differ? In a nutshell, we wanted to investigate the similarity of SCMs between subjects in addition to the similarity between subjects- and algorithm-based SCEs.

A caveat regarding the analysis and explanation of human judgements is that sample bias may distort conclusions. Sample bias has long been identified within the behavioral and social sciences as limiting the generalization of results obtained in a specific sample to the population. A common methodological fix to counteract such biases is to increase the sample size, see (Daniel, 2017) for a recent application and discussion. Certainly, the observed sample will affect the way the difference (to e.g. algorithm-based SCE) turns out to be, but then again our research question is *not* concerned with all possible human explanations, but any. Furthermore, we chose data sets that model very general examples and thus offer accessibility to the general population since no single person might be an expert. Ultimately, this way of designing our experiment, while not removing sample bias of course, renders the bias's qualitative effect onto our subsequent investigation negligible.

In the following we provide a discussion of several interesting and important insights discovered through the human user study. Nonetheless, it is important to note that our results like most modern day interpretations of human behavior are of conjectural nature – sensible, educated guesses essentially. During this discussion, we will point to specific aspects of the descriptive statistics displayed in appendix Fig.8. The actual human data is also being appended for the sake of completion (click on the following link to access the anonymized human data: https://anonymous.4open.science/r/Structural-Causal-Explanations-D0E7/Survey-Human-Data-Anonymized.pdf). The questionnaire contains four examples with two, three, three, and four variables (or concepts) respectively that are being visually depicted in addition to a concise textual description. We randomized the textual description of up to three variables across all examples for any randomly selected participant. Doing so, we allow for the randomized concept to reverse causal influence directions, thus, diminishing the bias of chance-selecting said causal direction – in a nutshell, this randomization scheme helps us in controlling for explanation variance (or leeway) of the subjects. Nonetheless, we still observed that for any variable pair $(X, Y)$ the meanings of $X$ and $Y$ themselves could be interpreted differently, which ultimately resulted in False Negatives regarding agreement i.e., people will disagree technically although they actually agree. To give a concrete example, consider the following: pre-condition in Example 2 can be interpreted as "the length of the medical history of a patient" (negative; increasing implies lower chance of recovery) opposed to "the state of well-being of a patient" (positive; increasing implies higher chance of recovery), thereby some subjects might choose $Z_1 \to R$ while others will choose $Z_2 \leftarrow R$ where $Z_i$ are the different explanations of the "pre-condition" concept (and $R$ denotes recovery), yet all subjects agree on an existing relation between the two variables: $Z_i \leftrightarrow R$. Also, some variables/concepts were more stable in their explanation variance. To give yet another specific example, altitude and temperature in Example 1 (appendix Fig.7) are stable concepts while the aforementioned pre-condition in Example 2 is unstable (due to its explanation variance/leeway). More importantly these different explanations due to the ambiguity inherent in language become visible within the statistics. To stay inline with the previous example, consider the medical example within appendix Fig.8 (second row, middle) and specifically consider the edges $T \to R$ and $Z \to R$. For the former relation the agreement between subjects is evident i.e., the majority of human subjects will select this edge. For the latter relation, we clearly see the

two previously discussed explanations that subjects employ during edge decision. I.e., for some subjects the edge between $Z$ and $R$ is positive and for some others it is negative, while naturally all agree upon there being a relation between the variable pair ($Z \leftrightarrow R$) opposed to there being no relation ($Z \not\leftrightarrow R$).

We observe a systematic approach and thereby non-random approach to edge-/structure-selection by the human operators, see any of the subplots within appendix Fig.8. Furthermore, there are only a few clusters even with increasing hypothesis space. Both the systematic manner and the tendency to common ground are evidence in support of the MMC hypothesis (MM ≡ SCM, Hyp.1) and its implied argument on "true" SCM information reachable from the overlapping MM-based SCMs or SCMs.

Although we randomize the order of variables in addition to consistently presenting them in a simple line with the intention of not inducing any specific sorting/structure to avoid bias, we still observed apparent, unintended subject behavior. For instance, subject number 5 only considered pairs presented next to each other as being questioned although the other combinations are meant to be queried as well. While additional research needs to corroborate these observations, our data suggests that attention might have decreased over the course of the experiment for a subset of subjects as suggested by e.g. subject number 7 where overall agreement with the subject majority is to be found but eventually at the very last example "mistakes" occur (specifically, the subject highlighted that "increasing age increases mobility", in stark disagreement with the majority of participants). We also observe that the increase in hypothesis/search space (i.e., more variables) comes with an increase in variance. This variance increase can be argued to be due to the progressive difficulty of inference problems as well as decreased levels of attention and potential fatigue across the duration of the experiment (e.g. consider the duplicate plots, third column, in appendix Fig.8 where the number of unique structures that are being identified increases significantly). Yet another interesting observation concerns the aspect of time, consider subject number 17 where there is a cycle between treatment and recovery where the subject likely thought in terms of "increasing treatment increases speed of recovery *which subsequently* feeds back into a decrease of treatment (since the individual is better off than before)" which seems like a valid inference but clearly considers the arrow of time. Yet another observation, some subjects faced questions of variable scope e.g. if there is a causal connection between food habits and mobility, then some subjects considered energy as the mediator and since energy is not part of the variable scope, confusion might arise whether to place an edge between food habits and mobility or not. In fact, for such a scenario the correct answer is to place an edge, since there exists a causal path from food habits to mobility, via energy, even if energy is not displayed. I.e., in causality, an edge can/will talk implicitly about all the more fine-grained variables that are part of the causal edge/path.

The second data set is an instance of the famous Kidney Stone example (Peters et al., 2017), where $Z$ is a confounder that indicates the pre-conditions in terms of e.g. the size of the kidney stone, and it also illustrates the famous Simpson paradox (Simpson, 1951; Pearl, 2009; Peters et al., 2017) where the recovery will favor one treatment in the overall statistics while being better for all of the non-consolidated views for the other treatment. We observe that not a single subject places the edge pre-condition to treatment ($Z \rightarrow T$) which is arguably at the core of Simpson's paradox. This observation gives an additional cue on why the phenomenon is called paradox because no human subject expects the existence of this connection and even actively neglect the existence.

We observe that the human-based SCE match the Ground Truth SCE *perfectly* up to the R data set SCE, which is also the "Result" in Fig.4 i.e., the "Mode" approach returns the correct SCE while the "Greedy" approach chooses the wrong edge type for $Z$ and $R$. After further investigation, we believe to have found several explanations for this "human" mistake that we discuss extensively in the appendix. On another note, we observe that the overall flawless performance of human-based SCE speaks for superiority over algorithmic graph learner-based SCE. To conclude this paragraph, let us appreciate one such drastic difference in explanations, which in fact occurred on our lead example "Causal Hans":

**Humans**: *"Hans's Mobility is bad because of his bad Health which is mostly due to his high Age, although his Food Habits are good."*
**Machines**: *"Hans's Mobility, in spite his high Age, is bad mostly because of his bad Health which is bad mostly due to his good Food Habits."*

A.9    OTHER NOTEWORTHY ASPECTS

A statement on the role of the MMC hypothesis. An algorithm for SCE-based graph learning. Technical details to the experiments. A reference to the remaining appendix figures that make the bulk of the remaining appendix.

**The Importance of MM $\equiv$ SCM for SCE.**    While the MMC is a fundamental question that cuts to the core of human thinking and remains to be proven right or wrong (although we believe it to be true to the extent of representability through SCM), and while we used it to ultimately justify the usage of SCM to then derive the causal explanations we call SCE, still, to the actual existence and formalism of SCE the MMC's truth value is invariant. Put blantly, if the MMC were to be wrong, then the formalism of SCE and all proven properties remain *the same*. However, if MMC were to be true, then SCE in fact become a "stronger" formalism for causal explanations since they'd have a direct link to the MM. More importantly, one could make the case that they'd represent a "natural" formal pendant to the vague human explanations.

**Simpson's Paradox Example.**    Consider the well-known Simpson's paradox example for the medical setting of Kidney stone treatments from (Charig et al., 1986). The setting is given by $T, K, R$ which are Treatment, Kidney Stone Size, and Recovery respectively, and further the graph is given by $T \rightarrow R, K \rightarrow \{T, R\}$. It is known that $T = 0$ denotes open surgery and $T = 1$ denotes Percutaneous nephrolithotomy (being a more involved procedure) and in the overall statistics for recovery of the patient (denoted by $R = 1$) we observe $78\%$ versus $83\%$ respectively, suggesting that $T = 1$ is the better option. Yet, when looking at the confounder $K$ values of patient recovery, we observe $93\%$ versus $87\%$ for a small kidney stone $K = 0$ and $73\%$ versus $69\%$ for a large kidney stone $K = 1$ respectively, suggesting that in fact $T = 0$ is better instead. This is the "paradoxical" situation, which is sensible from the *causal perspective*. If we now ask the single why-question for patient $i$ with say values $T = 1, R = 0, K = 1$ on why $i$ did not recover $r_i < \mu^R$ (where $\mu^R$ is the mean recovery of the data set), then we obtain an SCE that reads as follows: *"Patient $i$ did not recover because of the large kidney stone, although (s)he had Percutaneous nephrolithotomy."*

**Hidden Confounders in Semi-Markovian Models.**    As we pointed out in the main text, SCE can naturally handle/extend to semi-Markovian models. For illustration, consider the non-Markovian alternative to the example from the paragraph above on Simpson's paradox, where $K$ is a hidden confounder i.e., we only observe $T \rightarrow R$ as the graph. In common settings found in for example (Xia et al., 2021), we might at least be aware of the fact *that there is* hidden confounding present between the two variables and thus have an additional (dashed) bi-directed edge between $T$ and $R$ (case 1) and in the arguably worst case, said variable is fully undetected (case 2, in this case it is not necessarily a hidden confounder but simply a hidden cause, since we don't know if it is confounding or not—confounding meaning the same thing as *common cause*). Let's consider both cases, in case 1, the SCE for the same question as before would read as: *"Patient $i$ did not recover although (s)he had Percutaneous nephrolithotomy."* We note that simply the reasoning on $K$ is not being delivered, naturally, since $K$ is not in the SCM/CEM that the SCE process observes. For case 2, we'd observe the same reading due to the definition of the SCE construction. Here, however, we note that this case allows for a natural extension of SCE in which the reading could change to possibly, "Patient $i$ did not recover because of *an unknown reason*, although (s)he had Percutaneous nephrolithotomy." Note that this semi-Markovian SCE now allows for reasoning with "unknown reasons" since the hidden cause $K$ will certainly have a causal relation to $R$ (since $K$ is a cause) but the name of $K$ will not be revealed (since $K$ is hidden). With this example, we thus conclude that Markovianity can be leveraged by SCE.

**Algorithm for SCE Regularization.**    For sake of completion, we provide an explicit example algorithm for the simple penalty term we added in our setup for improving graph learning with available SCEs for the respective data set, consider the following:

---

**Algorithm 1** Graph Learning Using Arbitrary graph learner and SCE Regularization Penalty

---

**Input**: Data $\boldsymbol{D}$, Graph learner $\mathcal{M}_{\boldsymbol{\theta}}$, Optimizer $\mathcal{O}$
**Output**: SCM $H$, Causal Graph $G$

1: **while** $i \leq |\mathbf{I}|$ **do**
2:     $H, l \leftarrow \mathcal{M}_{\boldsymbol{\theta}}(\mathbf{D})$ {perform learning using available data}
3:     $Q_X, \boldsymbol{I}^* \leftarrow \boldsymbol{I}_i$ {acquire why question and corresponding SCE}
4:     $\boldsymbol{I} \leftarrow \mathbf{E}(Q_X, H, \boldsymbol{D})$ {estimate's SCE of graph estimate}
5:     $l_{\boldsymbol{I}} \leftarrow ||\hat{\boldsymbol{I}} - \boldsymbol{I}^*)||_2^2$ {compare estimate and ground truth}
6:     $\boldsymbol{\theta} \leftarrow \mathcal{O}(l, \boldsymbol{\theta})$ {parameter update using penalty}
7: **end while**
8: $H \leftarrow \mathcal{M}_{\boldsymbol{\theta}}(\mathbf{D})$, and $G \leftarrow |\tanh(H)|$ {return arguments}
9: **return** $H, G$

---

**Technical Details and Code.** *All experiments are being performed on a MacBook Pro (13-inch, 2020, Four Thunderbolt 3 ports) laptop running a 2,3 GHz Quad-Core Intel Core i7 CPU with a 16 GB 3733 MHz LPDDR4X RAM on time scales ranging from a few seconds (e.g. evaluating SCE in Exp.2) up to approximately an hour (e.g. SCE-based learning in Exp.3). Our code is available at: https://anonymous.4open.science/r/Structural-Causal-Explanations-D0E7/README.md.*

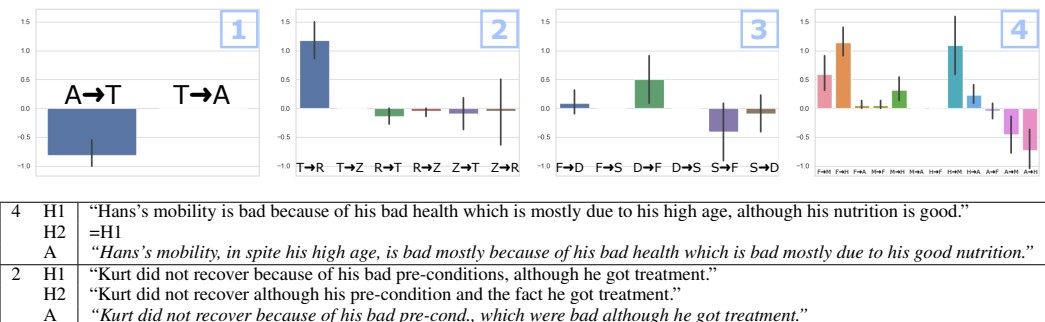

| 4 | H1 | "Hans's mobility is bad because of his bad health which is mostly due to his high age, although his nutrition is good." |
| | H2 | =H1 |
| | A | *"Hans's mobility, in spite his high age, is bad mostly because of his bad health which is bad mostly due to his good nutrition."* |
| 2 | H1 | "Kurt did not recover because of his bad pre-conditions, although he got treatment." |
| | H2 | "Kurt did not recover although his pre-condition and the fact he got treatment." |
| | A | *"Kurt did not recover because of his bad pre-cond., which were bad although he got treatment."* |

Table 3: **"Humans vs Machines".** Top: Edge plots per example where the bars denote the average value of given relation and the errors confidence intervals. Bottom: The SCE generated for the two human variants (from main paper Fig.4) against a graph learner representative (NT, Zheng et al. (2018)). Human explanations are (near-)identical to the ground truth from Tab.2. (Best viewed in color.)

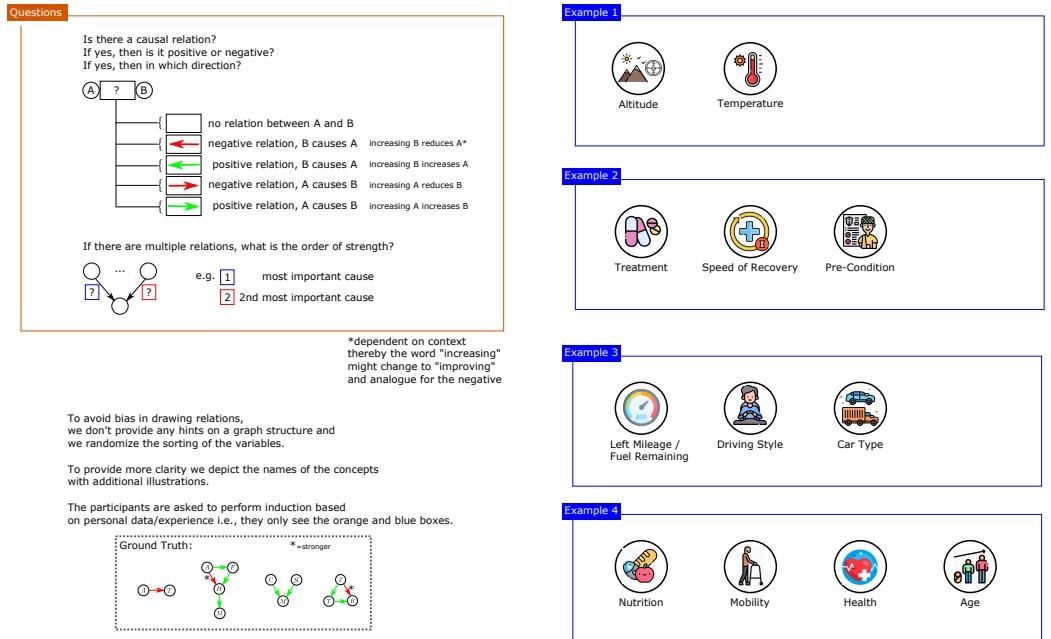

Figure 7: **Experiment Setup for the Human Case Study.** The participants are being asked two questions: whether there is a directed relation between some variable pair $A$ and $B$, and when there are multiple causes how they behave relatively i.e., the order of strength in relations. We avoid bias in drawing relations by randomizing the order and presenting the variables in a sequence. Induction is being performed from personal "data"/experience, rather than by looking at a matrix of data points. (Best viewed in color.)

| | Structural Causal Interpretation (Thm.1) |
|---|---|
| #1 | Dataset: DW, Query: "Why is the temperature at the Matterhorn low?" |
| GT | "The temperature at the Matterhorn is low because of the high altitude." |
| $\mathcal{M}_1$ | "The temperature at the Matterhorn is low because of the high altitude." |
| $\mathcal{M}_2$ | "The temperature at the Matterhorn is low because of the high altitude." |
| $\mathcal{M}_3$ | "No causal explanation for Matterhorn's temperature." |
| #2 | Dataset: DW, Query: "Why is the Matterhorn so high?" |
| GT | "No causal explanation for Matterhorn's altitude." |
| $\mathcal{M}_1$ | "No causal explanation for Matterhorn's altitude." |
| $\mathcal{M}_2$ | "No causal explanation for Matterhorn's altitude." |
| $\mathcal{M}_3$ | "The altitude of the Matterhorn is high because of the low temperature." |
| #3 | Dataset: CH, Query: "Why is Hans's mobility bad?" |
| GT | "Hans's mobility is bad because of his bad health which is mostly due to his high age, although his nutrition is good." |
| $\mathcal{M}_1$ | "Hans's mobility is bad because of his bad health which is bad because of high age and mostly due to his good nutrition." |
| $\mathcal{M}_2$ | "Hans's mobility is bad because of his good nutrition." |
| $\mathcal{M}_3$ | "No causal explanation for Hans's bad mobility." |
| #4 | Dataset: CH, Query: "Why is Hans old?" |
| GT | "No causal explanation for Hans being old." |
| $\mathcal{M}_1$ | "No causal explanation for Hans being old." |
| $\mathcal{M}_2$ | "No causal explanation for Hans being old." |
| $\mathcal{M}_3$ | "Hans is old because of his good nutrition and bad mobility, which is because of his bad health." |
| #5 | Dataset: CH, Query: "Why is Hans's nutrition good?" |
| GT | "Hans's nutrition is good because of being older." |
| $\mathcal{M}_1$ | "Hans's nutrition is good because of being older." |
| $\mathcal{M}_2$ | "No causal explanation for Hans's nutrition." |
| $\mathcal{M}_3$ | "Hans's nutrition is good because of his bad health and mobility." |
| #6 | Dataset: M, Query: "Why is your personal car's left mileage low?" |
| GT | "Your left mileage is low because of your small car and your bad driving style." |
| $\mathcal{M}_1$ | "Your left mileage is low mostly because of your small car and because of your bad driving style." |
| $\mathcal{M}_2$ | "No causal explanation for the left mileage." |
| $\mathcal{M}_3$ | "Your left mileage is low because of your small car and your bad driving style." |
| #7 | Dataset: M, Query: "Why is your personal car small?" |
| GT | "No causal explanation for the car size." |
| $\mathcal{M}_1$ | "No causal explanation for the car size." |
| $\mathcal{M}_2$ | "Your personal car's size is small because of your good driving style and fuel savings." |
| $\mathcal{M}_3$ | "No causal explanation for the car size." |
| #8 | Dataset: R, Query: "Why did Kurt not recover?" |
| GT | "Kurt did mostly not recover because of his bad pre-conditions, although he got treatment." |
| $\mathcal{M}_1$ | "Kurt did not recover because of his bad pre-conditions which is because of the treatment he got." |
| $\mathcal{M}_2$ | "No causal explanation for Kurt's recovery." |
| $\mathcal{M}_3$ | "No causal explanation for Kurt's recovery." |
| #9 | Dataset: R, Query: "Why did Kurt get treatment?" |
| GT | "Kurt got treatment because of his bad pre-conditions." |
| $\mathcal{M}_1$ | "No causal explanation for Kurt's received treatment." |
| $\mathcal{M}_2$ | "Kurt got treatment because of his bad pre-conditions." |
| $\mathcal{M}_3$ | "No causal explanation for Kurt's received treatment." |

Table 4: **More NIM-based SCE.** We prove Thm.1 for general NIM while pointing to some example methods from the existing literature on NIM. Here we show the results of running the methods $\mathcal{M}_i$, 1:NT (Zheng et al., 2018), 2:CGNN (Goudet et al., 2018), 3:DAG-GNN (Yu et al., 2019) on the four data sets weather forecast (W, Mooij et al. (2016)), health (H), mileage (M), and recovery (R, Charig et al. (1986)) for the respective queries. As suggested, the methods are explainable and reveal insights onto the learned causal semantics, while varying significantly in quality in terms of accuracy relative to the ground truth (GT). Independent of accuracy, "No causal explanation . . . " occur when the SCM estimate of $\mathcal{M}_i$ contains no causal path to the queried variable $X$ i.e., $\text{pa}_X = \emptyset$ (supported through GT sparsity). We also show GT explanations that require a negative "no answer" response by $\mathcal{M}_i$.

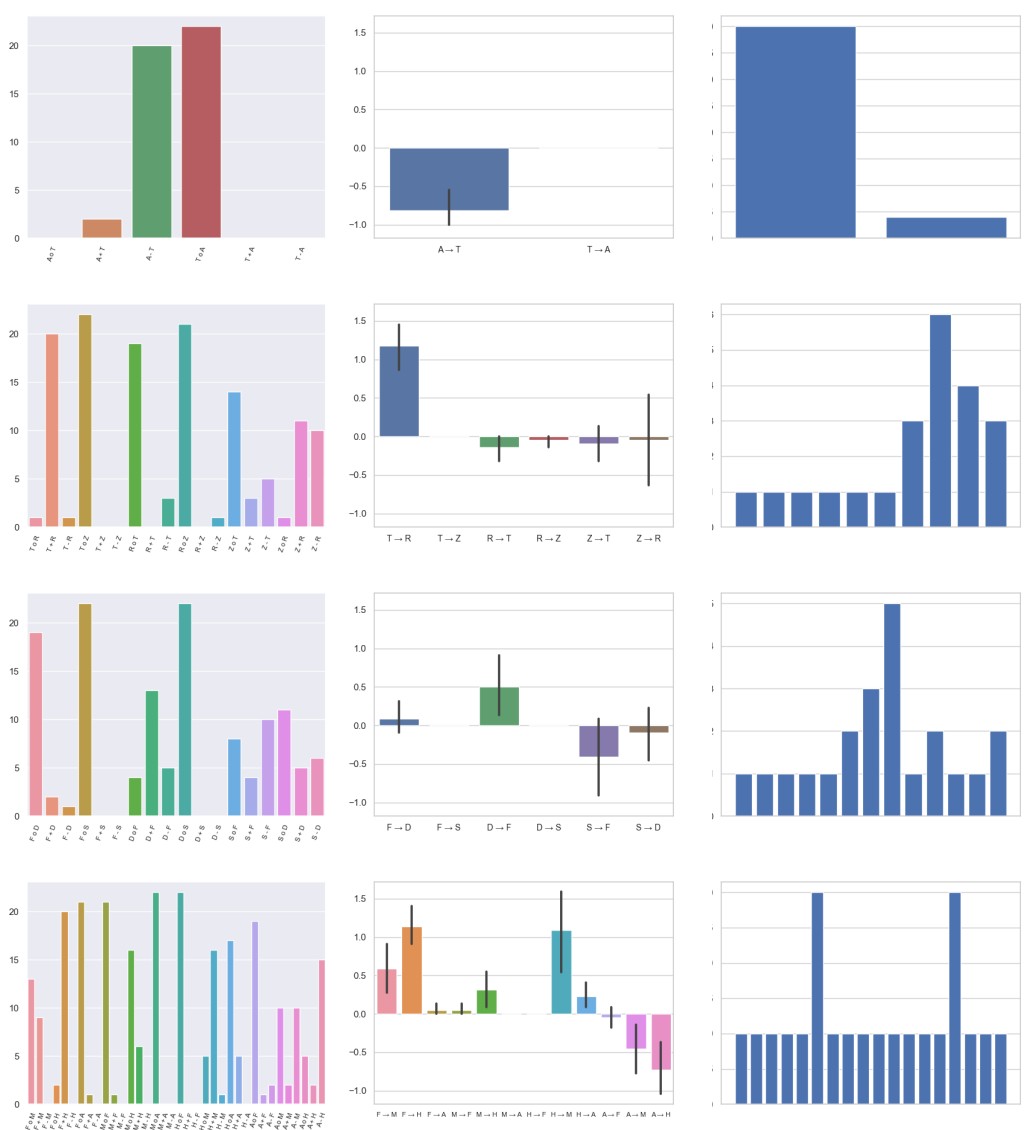

Figure 8: **Human Data Analysis: Qualitative, Quantitative, and Uniqueness.** Statistics collected from the human data ($N = 22$). The rows denote the four data sets: weather forecast (W, Mooij et al. (2016)), health (H), mileage (M), and recovery (R, Charig et al. (1986)). The columns: qualitative edge distributions that show for each of the different edge type how often it was chosen respectively (left), quantitative edge distribution for each edge where the error bars denote confidence intervals (middle), and the unique structure counts where each bar depicts the frequency of a qualitative structure discovered by the human subjects (right). Extensive elaboration on the setup, execution and results of this human study are to be found in the corresponding appendix section. (Best viewed in color.)

