# OpenReview forum: "Causal Explanations of Structural Causal Models"
_ICLR.cc/2023/Conference — Submitted to ICLR 2023_

### Official Review · Reviewer_Upiw · 2022-10-25

**Confidence:** 3
**Correctness:** 3
**Technical Novelty And Significance:** 3
**Empirical Novelty And Significance:** 2
**Recommendation:** 6

**Clarity, Quality, Novelty And Reproducibility:**

- The paper is easily readable.
- Aspects of the contribution exist in prior work. The contributions are only marginally significant.
- A GitHub repository is provided by the authors. Experiments in the paper are fully reproducible.

**Strength And Weaknesses:**

Strengths:
+ The paper is self-contained, and the contribution is well structured at the conceptual level. Thanks to straightforward examples and infographics, it is also clear for the reader to understand.
+ Related work is condensed but rather complete, and background discussion places the current work in a relevant and impactful study area.
+ Due to the approach's user-oriented nature, using a user study to evaluate the understanding of explanations is an engaging strategy.

Weaknesses:
- The availability of the full SCM is a strong assumption. In general, it is not testable and may thus not hold in practice.
- Authors should introduce a discussion on possible limitations of the algorithm
- All datasets used for the experimental evaluation contain a maximum of 4 variables. I would suggest to test the proposed approach w.r.t. dataset with higher dimensionality.



**Summary Of The Paper:**

This paper addresses the problem of deriving causal explanations for the decisions of opaque Machine Learning models.
Unlike current state-of-the-art methods, this work proposes an explanation method that aspires to consider causal interactions by exploiting the knowledge of a given SCM. In particular, the novelty of the approach relies on the capacity to distinguish between direct and indirect effects and the possibility of identifying information on the causal effect with positive and negative cases.

**Summary Of The Review:**

A well motivated paper with extensive and comprehensive experimentation. The evaluation of the paper would improve if the authors could be able to address the weak points highlighted above.

---

> ### Author Response · Authors · 2022-11-11
> **Response and Discussion**
>
> Thanks for the nice review! We were hoping that having the user study corroborate the presented user-based method would make sense and we are glad that you liked it!
>
> Indeed, the assumption with the SCM being available (even only partially) is strong. Though, the great thing is that we can (a) also only use a causal graph without parameters (so no SCM is strictly necessary) and (b) still use the method to improve when being given even a wrong causal graph as long as some parts are right about it. Then again, we can make the argument that the research question here is really about "Okay we have finally acquired some parts of the SCM, what now?", that is, it is not just important to discover SCMs but also use them subsequently.
>
> Condensing our discussions about limitations into an explicit section "limitations" is a great idea! We were not able to present it explicitly in the main text due to space constraints but note that we do point to limitations such as  SCE for hidden confounders (Pg 6, last paragraph) and the cognitive science aspects with the MM=SCM hypothesis (Pg 4, section 3.1, 1st paragraph and footnote 2) as we discuss throughout the paper. If accepted and thus given an extra page luxury, we will do this more explicitly.
>
> Yes, the low dimensionality is not ideal, but then again, this is something which plagues current causality research, and especially foundational research like this since we introduce a completely new way of thinking about causal explanations! Of course, extending SCE's to large dimensional data sets is an interesting immediate next step.
>
> Thanks for the great review! Looking forward to any further discussions with you!

---

### Official Review · Reviewer_j3te · 2022-10-25

**Confidence:** 4
**Correctness:** 3
**Technical Novelty And Significance:** 3
**Empirical Novelty And Significance:** 3
**Recommendation:** 8

**Clarity, Quality, Novelty And Reproducibility:**

The paper is clear and easy to follow. Aside from already mentioned parts that are exclusively in the Appendix. In particular, the toy example of Hans problem enables a clear instantiation of the proposed method and clarifies some difficulties in understanding.

There is no doubt that this article advances the understanding of explanations of machine learning methods by making proper assumptions (i.e., more artifacts than data alone) and by providing causal answers.

Reproducibility criteria are a strong point in this paper. The authors provided both the code and data. I did not have the opportunity to reproduce the results, but everything seemed fine.

Minor suggestions:

- Include (FOL) in the Parag. after Def. 2, since it is the first appearance of the First-Order Logic term.
- Fix sentence in Parag. 1/Sec.3.2.
- Fix "algoirthm" to "algorithm" in Proof of Theorem 1.
- Fix "the that the..." to "that the..." at the end of Sec. 4.1.
- Define NT in Sec. 4.2.

**Strength And Weaknesses:**

Strengths:

- This paper presents a new causal-based explanatory method. Since many learning algorithms produce black-box models (e.g., Deep Neural Networks), explanatory methods are needed, and the causality aspect of the proposed method turns its application more viable in real-world settings (e.g., decision-making in healthcare).
- When guiding the readers through a toy example, the authors provide the necessary intuition about their contribution.
- The tentative validation  (see weaknesses below) of the failure of previous methods to explain ML predictions is relevant to highlight the contribution of this paper.

Weaknesses:

- As far as I understood, the proposed method assumes the learned (from data) causal graph, not the SCM. For instance, in Sec. 4.2, the authors investigate the possibility of generating causal explanations from the learned causal graph.
	- Or is it the case that after learning the causal graph, their method learns the coefficients of a linear model to infer the equations of the SCM? (assuming linear structural causal equations)
In any of these possibilities (or other understanding), the authors should clarify this issue in the text.
- There is a lot of additional information in the Appendix, including important discussions, such as:
	- The relationship between the proposed method and Actual Causality.
	- Possible shortcuts to overcome the hidden confounder limitation of the proposed method.
There is no doubt that the authors provided a detailed Appendix for key points, and there is no sufficient space to include this information in the main text. My concern is that, at least, the above bullets should be in the main text.
- As discussed in Sec. 3.2, a limitation of the proposed method is that it doesn't consider hidden confounders. This assumption is unrealistic in several real-world scenarios.
- In Sec. 4.1., I did not understand the two shortcomings of CXPain. Since the SCM is given to both (i.e., SCE and CXPlain), it contains the information to derive the causal relations among the variables. Isn't it possible to use the input SCM to specify (i) direct/indirect causes among CXPlain's output? and  (ii) infer the causal effect among the output of the same explanation?
- In terms of limitations, I believe that would not be easy to learn the causal graph for high-dimension data  (e.g., images/videos) that maintain semantic information in the graph. Hence, if that is the case, the authors should highlight the discreteness requirement of datasets for which their method is suitable.
- Besides the cases of queries and explanations, I would like to see some general methodology to evaluate the quality of several queries. If that is not possible yet (i.e., it is an open research problem), the paper deserves some discussion on this.
- Given definitions and the analysis of theoretical parts of ER and SCE, it is still unclear how there are a loop works and how the input from decision-makers, for instance, may alter the final explanations.

**Summary Of The Paper:**

The focus of this paper is on Explanatory Interactive Learning (XIL), which consists of a loop in which the user queries the system, the system explains its predictions and the loop restarts.

An assumption here is that this interactive loop is interesting to both the system (improving its prediction accuracy) and the user (increasing their trust in the system).

The authors seek to causal XIL. Shortly the research question is how the ML system would provide causal explanations.

Additionally, they argue that previous explanation methods are not causal, even when they assume a Structural Causal Model (SCM). Their strong baseline is the CXPlain method, and they show that CXPlain doesn't provide causal explanations.

Hence, the authors propose an explanation method founded in a given SCM. They refer to their method as Structural Causal Explanation (SCE). Additionally, they conducted a user study to investigate the causal explanations based on SCMs.

Aiming to be self-contained, the paper should add some explanations on its relatedness to Actual Causality (mentioned at the end of Sec. 2).

It is a bit confusing when the authors mention M to be simultaneously (i) a valid why question and (ii) some proxy SCM (assuming that the hidden SCM is M^*).

**Summary Of The Review:**

This paper address a relevant topic, i.e. explanations of learned models, and do this by exploiting concepts and techniques from the Causal Inference literature. The authors provide a toy example that guides the readers throughout the paper. Additionally, they validate several hypothesis and provide detailed information in Appendix.

---

> ### Author Response · Authors · 2022-11-11
> **Response and Discussion**
>
> Thank you very much for this great review. Independent of the score which of course favors our paper, we are simply happy that it seems that all parts of the paper that we deem relevant and that we hoped would reach the community were recognized in this review!
>
> Regarding the question in the summary with $M$ and $\mathcal{M}$ note that it was the difference between M as in Mobility and M as in Structural Causal Model. Simple as that, but yes, having the same letter appear for the same concept can be overwhelming.
>
> For the prior knowledge part: in the XIL setting that we suggest we explore what we can do when we are given the causal graph and some coefficients (but coefficients are not necessary for the approach to still be sensible, it is just additional knowledge put to good use), and we do so because of the motivation in XIL that we can have an expert or simply some kind of basic expectation of how the thing we are investigating might work.
>
> You are definitely right, we should somehow squeeze in some of the key discussions of the appendix into the main text. While this might be very difficult as of now since arguably having the clear examples that you and the other reviewer enjoyed is more important, in the case of acceptance where we get an extra space, we can definitely add those!
>
> Yes, hidden confounders are a natural next step, very good and important point! Explanations for situations when we "don't know" so to say. While we explicitly say on page 6 that we leave this for future work, as arguably what we are doing is already foundational, we have a preliminary discussion in the appendix as you might have noted on page 20.
>
> Indeed, the observation with the CXPlain is exactly why it is a shortcoming in our opinion since we actually do provide everything for CXPlain and still get the bad results. Therefore, we can claim it to be a fair comparison since we provide the same knowledge and data to both methods and clearly see CXPlain suffer.
>
> Yes, very crucial point with the high-dimensional causal discovery! Unfortunately, we generally see this picture in the landscape of causal machine learning.. However, to be fair and protect our work a little, the discovery question is orthogonal to the question we are discussing here and so should not be used as an argument against what we present. But certainly, this is something which also harms our work that we struggle to acquire larger scale causal graphs.
>
> Very cool idea with the Causal XIL Loop Example. Maybe we can fit the idea with presenting the actual loop with an example somewhere in the appendix!
>
> Thank you again for the time and great review, much appreciated! Let's discuss more?

---

### Official Review · Reviewer_9oUq · 2022-10-25

**Confidence:** 4
**Correctness:** 1
**Technical Novelty And Significance:** 2
**Empirical Novelty And Significance:** 2
**Recommendation:** 3

**Clarity, Quality, Novelty And Reproducibility:**

The paper is mostly clear, except for the presentation of the empirical results. The paper appears to be novel.

**Strength And Weaknesses:**

Strengths:

This paper addresses an important problem, and is clearly written. I found the Causal Hans example to be particularly helpful in understanding the paper's motivations, the problem scope, and the approach taken.

Opportunties for Improvement:

Despite the fact the paper addresses an important problem and communicates the approach clearly, unfortunately I have many signficant concerns about this work.

(1) Theorem 1 is vague to the point of being unfalsifiable. To see this, consider an absurd structure learning algorithm that simply selects a random graph without looking at the data. While the statement is literally true that we could compute SCE, it says nothing about the quality of the derived explanation. Should a reader interpret the Theorem literally, in which case it is uninformative about explanation quality?

(2) If my interpretation of Theorem 1 is correct, and it is claiming that the output of a sufficiently well behaved structure learning algorithm (e.g. a sound and complete algorithm that returns the entire markov equivalence class of graphs), then the Theorem is incorrect. Graph structure learning algorithms can not return unique structural causal models, as many structural causal models map to the same causal graph. In other words, there is a type mismatch between the output of a structure learning algorithm (equivalence class of causal graphs) and the inputs of the SCE procedure (a unique SCM). See Bareinboim et al for discussion of the relationship between SCMs and graphs.

(3) The paper describes "why" queries in english language that map to formally to search procedures over counterfactuals. However, the authors do not take into account that counterfactuals involve holding exogenous noise fixed between factual and counterfactual worlds, nor do they consider the large literature on causal explanations with SCM. This is a bit perplexing, given that the authors cite Halpern 2016 as an analogous set of definitions.

(4) The empirical results are presented in a way that is extremely uninterpretable. For example, the rightmost column of Figure 8 has no labels or numeric values at all. It is very important to have (at least) a summary of the empirical findings in the main body. In fact, I do not see anywhere in the appendix a table or graph comparing the proposed approach with CXPlain on a well defined metric.

Minor concerns

Page 3 - "A great deal of research in causality (especially for ML) is concerned with leveraging observational data to reason about causal relatinoships (also known as identification), ..." This is not exactly right. The task would be better described as "effect estimation". Identification, or identifiability, is when one wants to assess whether their causal assumptions are necessary to yield unique causal conclusions.

Page 4 - "We call Q_X := R(x, \mu^X) a single why question if Q_X is true". This seems like a type mismatch. If Q_X is the relation, then it is always a question. Is the intention here to say that the why question Q_X is a function that maps from R to explanations or answers?

Bareinboim, Elias, et al. "On pearl’s hierarchy and the foundations of causal inference." Probabilistic and Causal Inference: The Works of Judea Pearl. 2022. 507-556.

**Summary Of The Paper:**

In this submission the authors present an end-to-end procedure for explanatory interactive learning. Most prominently, this submission focusses on a logical approach to constructing causal explanations of binary ordering relations, i.e. answers to "why" questions. They go on to test their approach in a survey of 22 participants.

**Summary Of The Review:**

The paper presents an interesting and important idea in end-to-end interactive causal explanations. However, I have significant concerns about the theoretical and empirical claims.

---

> ### Author Response · Authors · 2022-11-11
> **Response and Discussion**
>
> Thank you for the nice review. It is great to see that our technical writing is being recognized also with the highlighted examples. However, this of course comes at the cost of not being able to fit more in the main text.
>
> It is a very good point with Theorem 1 that you raise. It is actually the former point that you mention i.e., that you can simply compute the SCE and it is not a claim about any well-behaved aspects of learners. Formulating it in this way was our design choice to make it more reader understandable. Indeed, it is simply stating that it can be computed, which obviously turns out to be trivial, but since a theorem does not depend on any notion of how difficult it might be to be proven, we wanted to highlight this fact. Maybe we can simply opt for using something like "Key Insight 1". We definitely wanted to highlight the fact that it is applicable to any of such methods and since you have to prove that the requirements were fulfilled, we went with Theorem. But yes, something like Key Insight or Observation might be better here. This automatically also resolves point (2) that you mentioned as a hypothetical aspect being totally right about it.
>
> Indeed, we do not use counterfactuals, nor interventions for that matter, but actually the whole SCM or to be more precise the causal graph plus any other parameter estimates (in linear SCM for instance the coefficients). When we described the parallels to Halpern, what we meant was the way that Halpern creates the definitions in the first place. It is something foundational, Halpern is trying to formalize in a nice way things like actual causation, as you are correctly aware of, and we believe that our research is "foundational" in a similar way in that we also develop these rules etc. from ground up. That is why we tried to be as clear as possible in writing and in the examples, just like it is important for the understanding of concepts like with Halpern.
>
> Thanks for also checking out the appendix! We simply missed out on this. To be fair, the appendix should not be biting us back in evaluation and we believe you'd certainly agree here but we definitely will correct this. Regarding the well-defined metric for comparing CXPlain, well, that was our attempt actually of doing so since the whole setup around CXPlain and similar explanation methods is very different. To give a comprehensive analogy in terms of causality: it is like trying to compute counterfactuals in CBN (opposed to FBN) which is as you know, well, impossible since it cannot even be defined. On another note, regarding Fig.8, good catch, we simply forgot to add the axes and will do so.
>
> Thanks again for the great points! Probably (1) was the key aspect and that should be resolved with our proposal here. We hope that we answered your major concerns. Let's keep the discussion going, thank you!

---

### Official Review · Reviewer_vYZo · 2022-12-03

**Confidence:** 4
**Correctness:** 2
**Technical Novelty And Significance:** 2
**Empirical Novelty And Significance:** 1
**Recommendation:** 3

**Clarity, Quality, Novelty And Reproducibility:**

Clarity: As mentioned above, stronger contributions are listed in the introduction and abstract than in the technical contributions section after "we make several contributions:", so it is hard to determine exactly what the claimed contributions are. Furthermore, the listed contributions are vague; for instance, "(III) we apply the SCE algorithm [...]", "(IV) we discuss [...] how [...] [to] use SCE for improving model learning", and "(V) we perform a survey [...] to investigate the difference between user and algorithmic SCE" do not seem to be results in themselves and require the reader to guess what the authors' actions are demonstrating based on what has been stated in the abstract/introduction. The notation in Def 3 is unclear and hard to read. What are $R_1$ and $R_2$? It seems there should be an exists statement in ER1 and ER2 on $R_1$ and $R_2$. Why is $s$ necessary? Why is the output of $ERi(\cdot) \in \{-1, 0, 1\}$ rather than a binary output? Several other important technical points are unclear because it is not clear how the specifications in text translate to the notation (among others mentioned in the weaknesses section: is the paper examining individual explanations or global/SCM-level explanations, or something in between with population-level explanations?).

Quality: The content of the paper does not support its claimed contributions. See the weaknesses section.

Originality: The concept of an algorithm which recursively traces the ancestors of a variable which the algorithm is explaining to construct an explanation, as in Definition 4, is interesting and novel to my knowledge. The concept exists in Hall's notion of production but it hasn't been used to algorithmically construct explanations to my knowledge.

Reproducibility: The authors released their paper's code.

**Strength And Weaknesses:**

# Strengths

The paper's topic is very relevant to the field of explainable AI and the particular subfield of explanatory interactive learning (XIL).

The concept of an algorithm which recursively traces the ancestors of a variable which the algorithm is explaining to construct an explanation, as in Definition 4, is interesting and novel to my knowledge. The concept exists in Hall's notion of production but it hasn't been used to algorithmically construct explanations to my knowledge.

# Weaknesses

Overall, it seems that very strong claims are made in the introduction and abstract, and the claims are significantly weakened when the technical contributions are listed. Because these strong claims are not supported by the paper, they should not be present. I address these claims first, as they seem to be the strongest.

6. **SCE leads to improvements in (causal) XIL.** XIL is mentioned heavily in the abstract/introduction and in the discussion of Figure 1, which illustrates SCE being used in an XIL setup (a setup where users repeatedly query a system for explanations to better understand it). However, it is not shown in the paper that SCE actually leads to a better user experience in XIL. XIL and the setup in Figure 1 is not mentioned again in the body of the paper until section 4.1, where the authors argue CXPLain (the prior method) has shortcomings with respect to SCE, and in section 4.4, where the authors attempt to answer the question "What does SCE explain about the causal intuition that humans have that could provide for Causal XIL?". I do not see any mention of XIL in the arguments for SCE over CXPlain in section 4.1 or the analysis of the study in section 4.4. A human study would be useful to demonstrate this is the case.
7. **CXPlain does not output "truly causal" explanations.** Already, "truly causal" is a vague term which is not defined in the paper. Section 4.1 argues that SCE is superior to CXPlain because it provides additional information, but it is not shown that CXPlain's output is *wrong* or *not* "truly causal". In the example given (Figure 2), it seems reasonable to me to state that Hans' age is the main contributing factor to his (lack of) mobility, the explanation CXPlain seems to give, and the paper doesn't explain why this might be wrong. Thus, the claim that CXPlain does not output "truly causal" explanations doesn't seem to be supported in the paper.
8. **SCE explanations are human-understandable because they can be expressed using natural language.** First, I examine the claim that SCE explanations can be expressed using natural language. It seems that this is true for rules 1 and 2 (excitation and inhibition). However, rule 3 doesn't actually seem to correspond to the phrase "Y is *mostly* because of [...]"; it seems that it would be more accurate to say "[...] contributes the most to Y", and this may not constitute an explanation. For instance, it seems wrong to say "The amount of money a charity received was high mostly because of Alfred, who donated 2 dollars" while 500 other people each donated 1 dollar. So I conclude that SCE explanations can indeed be expressed using natural language, but these expressions may be misleading and may not align with what we intuitively consider an explanation. Further discussion of this point in the paper is warranted. Next, I examine the claim that SCE explanations are human-understandable due to their ability to be expressed using natural language. I could not find any content in the paper supporting this claim (including the human study in section 4.4, which did not show SCE explanations to humans); a human study should be used to support this claim.
9. **The paper argues in advocacy of Causal XIL.** I am unable to find any such arguments after reading the paper and searching for the term "XIL" in the paper in or before section 4.4, where the claim is made.
10. **Any causal graph learner can now provide human-readable explanations on any query of interest.** First, I examine the claim that this was not possible before (implied by the usage of the word "now"). I agree that causal graph learners' purpose is not to provide explanations. Next, I examine the claim that this is possible now. First, as discussed in point (8), there are issues with claiming that the generated explanations are now human-readable due to misalignment with human intuition. Second, it is not shown that explanations on larger graphs (for instance, 50-100 nodes) are human-readable. Third, causal graphs learners generally do not output a single graph but an equivalence class of graphs, and SCE doesn't operate on equivalence classes of graphs. Fourth, SCE is limited to queries comparing variables' values to their means (Def 1); this seems far from "any query of interest". Thus, there is still a large gap between causal discovery algorithms and human-readable explanations on any query of interest, which SCE does not bridge.

Next, I address the listed technical contributions. Overall, contributions 3-5 are not clearly stated, and their significance is unclear even with the context of the introduction and abstract. Contribution 1 requires more engagement with the prior literature on explanations and a clear statement + discussion of its limitations and assumptions. Contribution 2 relies on two qualitative arguments in section 4.1 which I do not think are sound.

1. **The structural causal explanations (SCE) algorithm + the claim that SCE explanations are "truly causal"** Regarding the construction of the SCE framework, there seems to be very little justification for Definitions 1-3 when taking into account prior literature.

    1.1. Regarding Def 1, why are "Why" questions constrained to variables' relationships with their means (Def 1)? Why aren't all kinds of why questions allowed, as in (Halpern, 2016)? It is stated after Explanation 1 that our "Why" questions are asked about *individuals*; if this is the case, are the unobserved variables in the SCM fixed?

    1.2. Regarding Def 2, how is each causal effect $\alpha_{X\to Y}$ specifically defined? Is this a unit-level causal effect specific to a setting of unobserved variables $\mathbf U$, or an average causal effect (a.k.a. average treatment effect)? If it is a unit-level causal effect, how is this supposed to be obtained, given that it is not possible to observe each variable's counterfactuals without the SCM? Is the assumption that we have the full SCM and the specific unobserved variable setting? If so, this should be stated explicitly in the contributions and the abstract, as it is even stronger than the assumption of just knowing the full SCM. And is the full SCM required? Or just the causal graph? If so, a causal graph is not sufficient to deduce all interventional effects from observational data, unless the graph is Markovian; if Markovianity is assumed, it should be explicitly stated as an assumption in the abstract/contribution section.

    1.3. Regarding Def 3., what is the intuitive justification for each rule? Examples in cases of linear SCMs are given. As mentioned in my comments on claim (8), we can obtain unintuitive behavior with rule 3 even in linear cases. I think we can also obtain unintuitive behavior with rules 1 and 2 if the causal effect of increasing & decreasing is on average 0, but there are specific regions of increase/decrease where there is an effect of increasing/decreasing the variable (e.g., a sine curve). How does this change the justification for each rule? If the assumption is that each function is linear and/or monotonic, this should be stated explicitly in the contributions and the abstract; in addition, justification for each rule should be given, as well as discussion of the rules' limitations.

    1.4. Regarding Def 4., it seems like a causal scenario $(\alpha_{X' \to Y'}, x', y', \mu_{x'}, \mu_{y'})$ is required for *every* pair of variables among the ancestors of the explained variable $X$. Is this correct? This isn't made clear in Def 2, and the causal scenario is referred to as a "single tuple", which makes it seem that there is only one causal scenario inputted to the SCE algorithm. In addition, why is the dataset $\mathbf D$ required for the SCE algorithm if the causal effects $\alpha_{Z \to X}$ are already known from the causal scenario? Is there some additional step of transforming a dataset to a causal effect which is not included among the definitions? And from what distribution (observational, interventional, etc.) is the data in the dataset drawn?

    1.5. Finally, it's not clear what "truly causal" means (as opposed to "causal" or "purportedly causal"), as this is not defined in the paper. The paper argues that SCE's outputs are "truly causal" by construction; it is not clear to me why this is the case.

2. **SCE fixes several of the shortcomings of previous explainers.** Section 4.1 argues that SCE is superior to CXPlain because 1) some aspects of the causal graphical structure (variables with direct vs indirect effects) cannot be distinguished using CXPlain's outputs and 2) CXPlain's explanation does not give information on how changing each input variable will change the outcome.

    2.1. The observation seems to be true, as CXPlain does not output any kind of recursive or graphical structure. But why is this a shortcoming? Is the explanation given by CXPlain wrong? Can't users look at the causal graph (which is required as an input to SCE) to obtain this information?

    2.2. This seems to be a valid point about CXPlain. But is it true that SCE does give this information in general? It is certainly true in linear cases, where the causal effect can be modelled with a single value. What about in non-linear cases, such as the sine wave case mentioned in my comments in (1.3)?

3. This seems to be an action the authors took rather than a result. Why are the applications mentioned in (3) novel/significant?

4. **Improving model learning using SCE.** It's not quite clear what "model learning" means here; this point should be clarified. If this refers to section 4.3, then there are some points to clarify. What is the exact definition of SCE regularization? How is error computed? Is the difference in Fig 3 statistically significant? Doesn't SCE already take in the true causal graph, so isn't including an SCE-based regularization equivalent to passing the graph learning method additional information on the graph (this of course depends on what the definition of SCE regularization is)? All of these question would need to be answered to assess the benefit of SCE regularization to causal discovery.

5. This seems to be an action the authors took rather than a result. Do the results of the study mentioned in (5) study support any novel/significant claims about SCE?

Actionable feedback:

1. Either remove claims 6-10 from the paper or include them in the listed contributions section and include supporting evidence for them in the body of the paper as suggested above. For instance, XIL seems interesting, but the connection to SCE doesn't seem to be made; it would strengthen the paper to make this connection. In general, please do not include strong claims in the introduction/abstract which are then weakened in the listed contributions section.
2. Remove claims 3-5 from the paper, as they do not seem to be contributions in themselves, and optionally replace them with novel/noteworthy conclusions from the listed application, discussion, and survey. Claim 5 could possibly be replaced with "analysis of a user survey of 22 participants on 4 explanation settings that explores challenges for explanation methods to tackle in the future".
3. Regarding claim 2, highlight specific cases where CXPlain fails at its stated goal. For instance, construct two examples where CXPlain outputs the same set of scores but SCE outputs different explanations. Conduct a human study to show that humans find explanations generated by SCE more useful for performing a task (or higher-rated according to some other criteria) than explanations generated by CXPlain.
4. Regarding claim 1, connect with the literature in greater depth to clearly state the assumptions/limitations of Definitions 1-3 in the abstract and contributions sections (or to change Def. 1-3 to encompass explanations as explored in prior work); some pointers include Halpern's approach to defining explanations and causation, which is cited in the paper (Halpern, 2016), and (MIller, 2017 - https://arxiv.org/abs/1706.07269). Clarify answers to the questions above in 1.1-1.4. Remove usage of the phrase "truly causal" or define it specifically.

**Summary Of The Paper:**

The authors list 5 technical contributions:
1. the structural causal explanations algorithm - "a new algorithm (SCE) for computing explanations from SCM making them truly causal explanations by construction"
2. an illustration of "how SCE fixes several of the shortcomings of previous explainers"
3. an application of "SCE algorithm to several popular causal inference methods" (I assume this is referring to the statement in the abstract, "Since SCEs make use of structural information, any causal graph learner can now provide human-readable explanations.")
4. a discussion "using a synthetic toy data set [of] how one could use SCE for improving model learning"
5. "a survey with 22 participants to investigate the difference between user and algorithmic SCE" (I assume this is referring to the statement in the abstract, "We conduct several experiments including a user study with 22 participants to investigate the virtue of SCE as causal explanations of SCMs.")

The authors also make the following claims in their abstract and introduction (and in the main body of the paper), which appear to be distinct from the claims listed in their technical contributions section:

6. SCE is a step forwards in terms of (causal) explanatory interactive learning (XIL). "In explanatory interactive learning (XIL) the user queries the learner, then the learner explains its answer to the user and finally the loop repeats. [...] Thus as a step towards causal XIL, we propose a solution to the lack of causal explanations."
7. CXPlain does not output "truly causal" explanations. "[...] we propose a solution to the lack of causal explanations. Specifically, we use the popular, proclaimed causal explanation method CXPlain to illustrate how the generated explanations leave open the question of truly causal explanations. [...] as we show in this work [CXPlain] still leave[s] open the question of truly causal explanations"
8. The explanations generated by SCE are human-understandable because they can be expressed using natural language. "We provide a new, natural language expressible (thus, human understandable) explanation algorithm with SCE."
9. "Throughout this paper we provided several arguments in advocacy of Causal XIL as the key paradigm of interest for future research and application."
10. By using SCE, any causal graph learner can provide human-readable explanations on any query of interest, and this was not possible before. "Since SCEs make use of structural information, any causal graph learner can now provide human-readable explanations. [...] [Theorem 1] tells us that any causal graph learner ever invented and that will ever be invented can provide causal explanations on any query of interest consistent with the learned model thus reflecting the learnt. [sic]"

**Summary Of The Review:**

While the paper is targeted towards an important field and introduces an interesting general concept of how to construct explanations, the content of the paper does not support its claimed contributions. The paper also needs reorganization to make its contributions clear, as it states stronger and distinct contributions in its introduction/abstract when compared to those that are listed in its contributions section. The paper also needs to clearly state the assumptions and limitations underlying its method, as described in the weaknesses section (my comments from 1.1.-1.4). Therefore, I recommend that the paper be rejected.

---

> ### Author Response · Authors · 2022-12-06
> **Response and Discussion (1)**
>
> Thank you for the extensive review! We appreciate the summary in list form (and the fact that you identified 10 of them), which eases reading but that also highlights both our main contributions and some "implicit" contributions.
>
> [First List] We will start rebutting the "burning" questions/points (6. to 10.) in the "Weaknesses" section, as you said yourself they seemed the most important to discuss:
>
> 6. While we never talk about improving user "experience", we believe you simply refer to the high-level idea/motivation for establishing our causal explanations framework in the first place, namely that Causal XIL would work with it (opposed to not working when using existing methods such as CXPlain, which is why we have one section discussing exactly that insufficiency of the status quo). Indeed, we don't have a walkthrough on a relevant real world example and having another human user study purely dedicated to this is surely a great idea for future work! However, we believe this point should not be held against us, as the paper is already packed to the brim quite literally and the main question at the end of the day is really how to establish the explanations in the first place, which we support extensively with the points you identified yourself.
> 7. You are right in saying that we never define "truly causal", but as becomes clear by reading the paper (and we believe you are aware of this, still we are highlighting it) "truly causal" is a description that refers to a anything that makes sensible use of a Structural Causal Model as in the Pearlian notion to causality (Pearl, 2009) be it explicit or implicit. Accepting that and looking back at CXPlain, we clearly see that it is not a truly causal method (in fact, it is stated to use Granger causality as inspiration explicitly by the respective authors) and we simply tried going the extra mile by showing the reader of our paper that even when applying it, it becomes apparent that the output is not sensible for what we are trying to achieve.
> 8. This one is actually very subtle, so bear with us. Great work on identifying that Rule 3 is somewhat different than Rules 1 and 2 and that it would be more something like "contributing most to". However, and here comes the important yet subtle part, Rule 3 *never* fires alone! It is a kind of modifier to Rules 1 and 2, which imply a "because"/"although" explanation, and then if Rule 3 fires as well you end up with something like "mostly because". We were already considering of renaming Rule 3 and following your observation, it most certainly seems like a sensible thing to do. Regarding the natural language implying human-understandable part, this one indeed we take for granted, as we never experienced any example which was not understood by us human developers and we'd still continue on arguing that this is for granted in our setup as the domains, the corresponding causal graphs (or SCMs) and finally the mechanism are all "human-understandable", but then again, we have to respect the fact that we are not providing a cognitive scientist's or psychologist's treatise with this submission, so maybe our view on this is restricted. If so, we'd love to see a counterexample.
> 9. This point is a duplicate of point 6. Therefore, consider revisiting what was written above to 6. as well. In any case, while the experiments 4.1 with existing "causal" explanation methods and 4.4 with human users should be in support of the suitability of SCE for Causal XIL, we believe that experiment 4.3 on using existing explanations for learning (Sec.4.3 and Fig.3) is the closest experiment to a valid walkthrough supporting the overall vision/motivation of Causal XIL that we provide at the beginning of the paper since it shows how learning the graph representations can be boosted by having our explanations readily available.
>
> **Continuation Follows**

---

> > ### Author Response · Authors · 2022-12-06
> > **Response and Discussion (2)**
> >
> > 10. Thanks for recognizing and agreeing that graph learners are not meant to provide explanations. The first sub-point we hope to have answered satisfactory in point 8. above. The second sub-point we did not show indeed, however, we can tell from our experiments that we conducted that this works without an issue, it is just the case that (as expected) the explanations (just like the graphs) become bigger. Very interesting though is then the question of attention, since the (human) user might not be interested in the whole answer but only parts of it. For the third sub-point we have to disagree, indeed, classical methods based on conditional independences will output for instance a Markov Equivalence Class but more modern methods based on functional assumptions or continuous optimization will (as presented in our experiments 4.2 on the quality of learned explanations) always output a single graph. Furthermore, this would anyhow not be an issue since SCE just needs any graph and you can simply pick any of the members of a given equivalence class. On another note, it is worthwhile remembering that SCE is an approach that generates causal explanations but whether the input graph that is being used for constructing those is actually the causal graph of the data you observe are two independent problems. Regarding the final, fourth sub-point, correct, we defined our question over "mean" comparisons, however, we could've generalized this to at least similar statistics of interest (say, the median) but since our experiments sufficed to using the mean, we stuck with it. Nonetheless, even for greater generalizations we believe them to be possible as the intuition about this part of the work is that we simply need a "population relative measure", that is, being able to compare an individual somehow against a population. Since we did not have any further results on this, we did not claim it.
> >
> > [Second List] Next we discuss the points from the second list regarding the technical contributions:
> >
> > 1. Since this point is going at each of the components of our derivation from Sec.3.1 we are going to look into each of them one-by-one. Just for clarification, indeed, this derivation (to the best of our knowledge) is the very first in the literature and that is why we designed Sec.3.1 as a "tag-along" derivation to really bring home the intuition. We believe what you mean in the separate points is the "additional" justification of the ideas in the context of prior work, so we will be sensible in answering following this thought.
> >
> >    Regarding sub-point 1.1. we answered in the fourth point of point 10. in the previous list that this restriction we mainly had since our results were restricted to it, but it is not necessary. Regarding the unobserved variables in the SCM, indeed they are fixed since the data that is given for the whole setup is a single data set and that implies that the SCM that generated this data fixed the unobserved terms. Note however that the SCE setup actually does not use knowledge on the complete SCM and therefore not unobserved terms, they are implicit for the observed data though.
> >
> >    Regarding sub-point 1.2. and the SCM, this we just answered in the previous sub-point discussion and the answer is No, we don't use nor require the full SCM. Regarding the causal effects, in our setting we often times look at linear SCMs as well which then equates the causal effects to simply the coefficients/weights in the linear equations but apart from that (as we stated in the paper) we consider typically attainable causal effects such as the ATE that you mention. Indeed, unit-level effects would be theoretically interesting and therefore reasonable to pursue for developing theoretical insight, but as you said yourself the incomputability is certainly a severe limitation for anything practical. Finally, regarding the Markovianity the answer is No, we do not necessarily assume Markovianity. We do look at Markovian SCM in our main examples but we also explore Non-Markovian SCM in a discussion in Appendix A.9. on page 20.
> >
> > **Continuation Follows**

---

> > > ### Author Response · Authors · 2022-12-06
> > > **Response and Discussion (3)**
> > >
> > > 1. (point 1. continued from "Response and Discussion (2)") Regarding sub-point 1.3. the intuition (as you would expect) is really the "tag-along" derivation we provide in all of Sec.3.1. with the Causal Hans example, however, to further strengthen the intuition of the reader (since we expect it to be difficult for learning about this newly introduced concepts) we provided discussions on the rules's properties in A.5. (p.15) and the mental model representable through SCM discussion (A.3, p.13) and implications (A.4, p.14) in the appendix sections. Thank you for this very interesting example with a sine-like curve! Indeed, we haven't discussed such a thing yet and it is certainly interesting as it is some sort of "context-specific" SCE. However, we believe it is fair to keep such an advanced topic as future work. Regarding the related question of monotonicity as possibly a required assumption that we missed out on, this would similarly require a deeper analysis. But thanks for raising these points, we can still be safe in the sense that all our examples further hold.
> > >
> > >    Regarding sub-point 1.4. thanks a lot for raising this point as we did realize by it that this part can be made more clear, you are correct that indeed we are talking about causal scenarios for each $(X,Y)$ pair along the ancestors of the explained variable. We see that using "$C_{XY}$ denote a causal scenario" was the point of confusion, for us it was clear since the rules always talk about any single $(X,Y)$ pair so technically it is correct, but it is difficult for a first time reader. We can very easily rewrite this. Regarding the data set $\mathbf{D}$ dependence, the answer is No, there is nothing hidden that you would have not expected. The $\mathbf{D}$ is simply "lazy" notation for avoiding to explicitly write out the causal effects and the Causal Scenarios subsequently for Eq.1 and of course it is all (as expected) just simple observational data (except the causal effect estimates of course like coefficients for linear or ATE from population level).
> > >
> > >    Regarding sub-point 1.5. this a duplicate of point 7. from the previous "hot topics" list above. Regarding SCE now specifically, they do make use of SCM knowledge (even if not necessarily fully as we observed in our discussion on unobserved variables or simply the fact that they don't require/use mechanisms but only some instance of a causal effect) and thereby they are causal by construction and accepting for the moment that Pearlian causal equals "truly causal" for this work, we can safely also conclude that SCE are truly causal. However, of course, if Pearl's notion to causality is not what "true causation" is about, then this argument does not work. However, this question is of purely philosophical nature and certainly out-of-scope here and practically irrelevant.
> > > 2. Since it seems to be about all relevant types of questions regarding CXPlain as one of the current popular status quo methods and its insufficiency for being causal and explanations relevant to Causal XIL, we (1) highlight the relevant sections of the paper being the discussion in Sec.4.1. p.7 with Fig.2 and Appendix A.2. p.10, and (2) we again summarize the 4 main shortcomings we could find. Those were: (1) CXPlain has no way of separating direct from indirect causes or any graphical intuition for that matter (as it simply cannot process a graph, so we provide the method with everything SCE can use, but if the method CXPlain cannot make use of - for whatever reason - then this certainly is a drawback at least in causal terms as there cannot be a guarantee towards the underlying SCM), (2) CXPlain has no way of differentiating positive from negative causal effects, (3) CXPlain has no way of discerning patients of same records (that differ only in unobserved variables), (4) CXPlain has no way of talking qualitatively about, say, age and mobility relation as for even similar patients the results can look drastically different. Disclaimer, we are not here to "bash" CXPlain but rather showed that the explanations it provides are simply not causal in the sense that we require, especially as we motivate for Causal XIL. We believe CXPlain has great place in the community and has stimulated research in this area, including this submission of ours.
> > > 3. Indeed, it is more of an "action" at first, but it is linked to our statement in Thm.1 p.6 which asserts the explainability of causal structure learning algorithms using SCEs, and in any case we wanted explicitly to highlight this observation as it empowers existing methods in the community, telling them that they can use their methods (assuming the output graph to be "sufficiently" causal in the sense that it at least partially captures the underlying data generating SCM) for computing explanations.
> > >
> > > **Continuation Follows**

---

> > > > ### Author Response · Authors · 2022-12-06
> > > > **Response and Discussion (4)**
> > > >
> > > > 4. Model learning in this section (and yes, it is Sec.4.3 with Fig.3) refers to learning graphs using any of the methods that you can also use for computing SCE. Indeed, we assume explanations existing from the correct graph (note however we could've used partially correct graphs in this setup equally well, expectedly we would likely see weaker results) and then simply use a naïve setup where we simply add a regularization term that based on some sensible metric for binary codes (here we used Structural Hamming Distance) can give us a penalty if the current predicted graph is not consistent with the implied graph by the explanation. The error is naturally computed using standard metrics deployed by such graph learners, such as count accuracies using true/false positive and true/false negatives. While we tested on 140 graphs in total, we did not compute whether the result is statistically significant. We believe that both the overall setup and the larger scale are indicative of that, but thanks for reminding us of doing a simple additional check. In any case, we leave an extensive exploration of this very interesting sub-direction of our work for future work, since there have arguably been not many works on using existing explanations for feedbacking learning.
> > > > 5. There are, as you would expect, multiple reasons for us conducting a user study. We explicitly already mention them throughout, see Sec.4.4 and even more so all extended discussions in Appendices A.3, A.4, and A.8. Also, opposed to point 3. which seemed like an action at first, we believe this point to certainly be a result right away. The reasons once again: (1) we could investigate our discussion on mental models being representable by SCM and the quality of modelling of SCM by humans, (2) we could compute the algorithm baselines against the human generated explanations confirming our intuition or rather initial hypothesis that humans outperfrom algorithms in terms of explanations generated (in that human models will have SCE that are a lot more appealing to what we deem correct), (3) to support the Causal XIL argument which assumes or expects a (human) user expert that can provide some intuition on the problem (in the sense of the underlying SCM) on the representation level of explanations (that is, SCE).
> > > >
> > > > Now, regarding the "actionable feedback" section, we simply provide an overarching comment covering points 1. to 4.:
> > > >
> > > > The actionable feedback covers mostly removing of certain claims or revising of certain claims as discussed above, since we believe to have answered all of the relevant points we would think that this actionable feedback becomes obsolete up to the points that we acknowledged to revise or improve. Some of these points were the analysis of the monotonicity assumption, the clarification of the confusion regarding the writing of the definition with causal scenarios or for instance computing a score of statistical significance for the graph representation learning experiment, which are very easily compiled for a final version.
> > > >
> > > > Next we mention a comment for the "Strengths" section. While we are very grateful for all these comments and believe this to be a great review challenging our paper and letting us identify solutions (as our engagement above and the arguably minor things that we have settled to add for a final version clearly suggest), it is somewhat odd that in this section you only mentions the topic being relevant and the recursive approach to computing SCE since again (independent of this initial rejecting score of the review) your review is very detailed and covering all relevant aspects of the paper. Nonetheless, thanks for acknowledging at least these two points as strongholds.
> > > >
> > > > **Continuation Follows**

---

> > > > > ### Author Response · Authors · 2022-12-06
> > > > > **Response and Discussion (5)**
> > > > >
> > > > > Finally, we answer and rebut the final questions that sneaked into the "Clarity, Quality, Novelty and Reproducibility" section:
> > > > >
> > > > > The only part with questions was the "Clarity" part, so we answer. For the part on Def.3 is a duplicate of point 1. in the second list of this rebuttal. As stated in the respective definition Def.3, $R_1$ and $R_2$ are the instantiations of the order relation ($<$ or $>$ in this case) and depending on the instantiation you might end up with say $ER1$ if it triggers explaining "because of high $X$" or "because of low $X$", so it acts as a modifier of the resulting explanation differentiating the different cases of explanations. Indeed, $s$ in Def.3 is not "necessary" but for ease of reading and also to make the assumptions/computation more explicit we put it there, because it highlights the fact that the rules make qualitative statements using the causal effects (so just needing the sign, whether it is positive/negative) opposed to quantiative statements (like causal effect being exactly a certain value $a$). For the $ERi$ output being non-binary, that is because of what just mentioned with the instances of $R_1,R_2$ and either $X$ contributing to $Y$ via a positive (say, $1$) and negative (say, $-1$) effect, however, choosing either to be $1$ or $-1$ is simply an irrelevant permutation as long as it is consistent throughout the rules.
> > > > >
> > > > > **Concluding Remarks** to this extensive rebuttal to a (very nice!) and extensive review. We politely disagree with the review on the parts that were claimed to be unsupported w.r.t. the contributions, as we discussed in the different sections above extensively, and since this was the main ground for rejecting our paper we hope that we have convinced you to revise your decision. Thank you again for your time and this detailed review!
> > > > >
> > > > > (This was the final part of the lengthy response, (5), looking forward to any further discussion!)

---

### Author Response · Authors · 2022-11-16
**Participation in discussion**

Dear reviewers,

We hope to have answered your concerns in our individual responses to the provided reviews. We would be happy to answer any further queries you might have before the end of the discussion period.
Do let us know if you found our response satisfactory or/and wish to take forward the discussion.

Regards,

The Authors

---

### Decision · Program_Chairs · 2023-01-20

**Decision:**

Reject

**Justification For Why Not Higher Score:**

The paper contains intriguing ideas but is not ready yet for publication due to some technical issues. For more details, please refer to the reviews and the subsequent discussions.

**Justification For Why Not Lower Score:**

N/A

**Metareview: Summary, Strengths And Weaknesses:**

The paper develops a new end-to-end procedure for explanatory interactive learning based on causal reasoning. The paper’s proposal was received in mixed ways by the reviewers. On the one hand, reviewers Upiw and j3te considered the method conceptually exciting and the idea of pursuing human validation compelling. On the other hand, reviewers 9oUq and vYZo raised some issues related to the technical and conceptual claims made by the paper. The technical quality and precision of the last two reviewers were high, and I appreciated reading their reviews and engagement.

After all, both sets of reviewers raised valid points — the paper has novel and compelling ideas but, at the same time, has serious technical concerns that need to be addressed before publication. Given the interesting ideas behind the paper, I would like to encourage the authors to take the detailed feedback provided into account and improve the presentation and clarity in its next iteration.